



# Retrieving the atmospheric concentrations of carbon dioxide and methane from the European Copernicus CO2M satellite mission using artificial neural networks

Maximilian Reuter[1], Michael Hilker[1], Stefan Noël[1], Antonio Di Noia[1], Michael Weimer[1], Oliver Schneising[1], Michael Buchwitz[1], Heinrich Bovensmann[1], John P. Burrows[1], Hartmut Bösch[1], and Ruediger Lang[2]

[1]Institute of Environmental Physics, University of Bremen, FB 1, P.O. Box 330440, 28334 Bremen, Germany
[2]EUMETSAT, Eumetsat Allee 1, 64295 Darmstadt, Germany

**Correspondence:** Maximilian Reuter (mail@maxreuter.org)

**Abstract.** Carbon dioxide ($CO_2$) and methane ($CH_4$) are the most important anthropogenic greenhouse gases and the main drivers of climate change. Monitoring their concentrations from space helps to detect and quantify anthropogenic emissions, supporting the mitigation efforts urgently needed to meet the primary objective of the United Nations Framework Convention on Climate Change (UNFCCC) Paris Agreement to limit the global average temperature increase to well below 2°C

above pre-industrial levels. In addition, satellite observations can be used to quantify natural sources and sinks improving our understanding of the carbon cycle. Advancing these goals is the motivation for the European Copernicus $CO_2$ monitoring mission CO2M. The necessary accuracy and precision requirements for the measured quantities XCO2 and XCH4 (the column-averaged dry-air mixing ratios of $CO_2$ and $CH_4$) are demanding. According to the CO2M mission requirements, the spatial and temporal variability of the systematic errors of XCO2 and XCH4 shall not exceed 0.5 ppm and 5 ppb, respectively.

The stochastic errors due to instrument noise shall not exceed 0.7 ppm for XCO2 and 10 ppb for XCH4. Conventional so-called full-physics algorithms for retrieving XCO2 and/or XCH4 from satellite-based measurements of reflected solar radiation are typically computationally intensive and still usually require empirical bias corrections based on supervised machine learning methods. Here we present the retrieval algorithm NRG-CO2M (Neural networks for Remote sensing of Greenhouse gases from CO2M), which derives XCO2 and XCH4 from CO2M radiance measurements with minimal computational effort using artifi-

cial neural networks (ANNs). Since CO2M will not be launched until 2026, our study is based on simulated measurements over land surfaces from a comprehensive observing system simulation experiment (OSSE). We employ a hybrid learning approach that combines advantages of simulation-based and measurement-based training data to ensure coverage of a wide range of XCO2 and XCH4 values making the training data also representative of future concentrations. The algorithm's postprocessing is designed to achieve a high data yield of about 80% of all cloud-free soundings. The spatio-temporal systematic errors of

XCO2 and XCH4 amount 0.44 ppm and 2.45 ppb, respectively. The average single sounding precision is 0.41 ppm for XCO2 and 2.74 ppb for XCH4. Therefore, the presented retrieval method has the potential to meet the demanding CO2M mission requirements for XCO2 and XCH4.



## 1 Introduction

Carbon dioxide ($CO_2$) and methane ($CH_4$) are the most important anthropogenic greenhouse gases because they are the main

drivers of climate change. Monitoring their concentrations from space is essential to identify and quantify anthropogenic emissions, thereby supporting the mitigation efforts needed to achieve the primary objective of the Paris Agreement of the United Nations Framework Convention on Climate Change (UNFCCC) to limit the global average temperature increase to well below 2°C above pre-industrial levels (UNFCCC, 2015). In addition, satellite observations can be used to study natural sources and sinks of these gases, contributing to a better understanding of the carbon cycle and thus improving climate predictions.

Advancing these goals is the motivation for the European Copernicus $CO_2$ monitoring mission CO2M (Meijer et al., 2020; Lespinas et al., 2020; Sierk et al., 2021), which will serve as a central element of the Monitoring and Verification Support (MVS) service capacity currently being developed as an integral part of the Copernicus Atmosphere Monitoring Service (CAMS). The mission involves the deployment of a constellation of three satellites, with the launch of the first CO2M satellite planned for 2026. CO2M builds on the heritage of the CarbonSat concept (Bovensmann et al., 2010; Velazco et al., 2011;

Buchwitz et al., 2013; Broquet et al., 2018).

However, the accuracy and precision requirements for the measured quantities XCO2 and XCH4 (the column-averaged dry-air mixing ratios of $CO_2$ and $CH_4$) are demanding and achieving them is a major scientific challenge. Specifically, the mission requirements document (MRD, Meijer et al., 2020) defines that the systematic errors of XCO2 and XCH4 shall not exceed a maximum spatial and temporal variability of 0.5 ppm and 5 ppb respectively. The stochastic errors due instrument noise shall

not exceed 0.7 ppm for XCO2 and 10 ppb for XCH4 for a reference scenario over vegetation. This is why CO2M is equipped not only with the main instrument CO2I ($CO_2$ Imager), comprising four imaging spectrometers, but also with the instruments MAP (Multi-Angle Polarimeter), which helps to better account for light scattering on aerosols and the BRDF (bidirectional reflectance distribution function), and CLIM (Cloud Imager), which helps to identify clouds in the field of view.

Conventional so-called full-physics algorithms for retrieving XCO2 or XCH4 (XGAS) from satellite-based measurements

of reflected solar radiation in the near-infrared (NIR) and shortwave-infrared (SWIR) spectral region require accurate radiative transfer (RT) and instrument simulations which are typically computationally expensive. Examples of such retrieval methods are described in the publications of Reuter et al. (2010, 2011, 2017b, a); Boesch and Di Noia (2023); Noël et al. (2021, 2022); Kiel et al. (2019); Guerlet et al. (2013); Cogan et al. (2012). Three full-physics algorithms for the analysis of CO2M data are currently also being implemented in the EUMETSAT ground segment. One of these methods is the Fast atmOspheric traCe

gAs retrievaL (FOCAL, Noël et al., 2024). The others are RemoTAP (Lu et al., 2022) and FUSIONAL-P, a further development of the algorithm described by Boesch and Di Noia (2023). It is anticipated that continuous analysis of the data stream from a single CO2M satellite using these three methods will require the computing power of several thousand CPU cores, and re-processing the data from two or more CO2M satellites will require several times that amount.

Despite the high computing power required, there are still a number of reasons that can lead to more or less large systematic

inaccuracies in the retrieved XGAS quantities. These can be simplifications of the RT (e.g. neglect of Raman scattering, neglect of polarization, reduced number of streams, reduced accuracy of scattering phase functions, 3D effects), which are necessary





to keep the computation time within acceptable limits. But also insufficiently characterized geophysical input parameters (e.g. spectroscopy, aerosol and cloud microphysical properties, BRDF, all kinds of subpixel inhomogeneities) and insufficiently characterized instrument properties (e.g. incomplete stray light correction, crosstalk or sensor nonlinearity) can play a role.

For these reasons, currently existing full-physics retrievals typically exploit more or less complex empirical bias corrections in order to meet the demanding accuracy requirements. This applies to established methods for instruments such as OCO-2 (Orbiting Carbon Observatory-2), GOSAT (Greenhouse Gases Observing Satellite), and GOSAT-2 (Reuter et al., 2017b, a; Kiel et al., 2019; Noël et al., 2021, 2022; Boesch and Di Noia, 2023; Guerlet et al., 2013; Cogan et al., 2012) and it is not unlikely that the same will apply to the CO2M XGAS retrieval algorithms currently being implemented by EUMETSAT, once

they are confronted with actual measurements.

The variance of the bias correction can be of the same order of magnitude as the retrieval increment, i.e., the difference between a prior knowledge and the result (Reuter et al., 2017a; Kiel et al., 2019), implying that the bias correction contributes a non-negligible fraction of the information of the result.

Most bias correction methods are empirical and usually based on supervised machine learning techniques. These include

simple multidimensional linear regressions (Kiel et al., 2019) or more complex methods based on, e.g., random forest regressors (Noël et al., 2022; Schneising et al., 2019, 2023). For this reason, they also face the issues associated with data-driven methods, such as the need for a representative training data set including ground truth.

Consequently one motivation for this study is to try to avoid the complicated and computationally intensive step of full-physics algorithms and instead analyze the measured spectra from the outset using a data-driven method. Multilayer perceptrons

(MLPs) are artificial neural networks (ANNs) that are well suited for this task and, once trained, can analyze large amounts of data with minimal computational effort. Basically, an MLP is a nonlinear function whose parameters are adjusted during training to best map the input features (e.g., spectra, meteorology, angles) to the output target (e.g., XCO2, XCH4). This is called supervised learning, and it requires a representative set of input features for which one or more known output target variables exist. The principle of the method is analogous to that of linear regression, which is one of the simplest forms of

supervised learning. A general introduction to MLPs can be found, e.g., in the textbook of Rojas (1996).

As is known from other regressors with many free fit parameters, MLPs tend to be good interpolators but poor extrapolators Krasnopolsky and Schiller (2003). This is particularly relevant because $CO_2$ and $CH_4$ increase over time, and a training data set consisting of today's measurements is not representative of the future. Furthermore, MLPs can learn from spurious correlations just as efficiently as from actual physical relationships, i.e., they can give significant weight to input/target correlations that are

not directly caused by a physical relationship, but by factors such as similar seasonality (e.g., XCO2 and solar zenith angle). However, generalized learning occurs only in the latter case, and applying the MLP to unknown scenarios leads to accurate data products only in this case. Another potential hurdle is that MLPs can be affected by uncertainties in the training target. Consequently, it is necessary to ensure that the training data set is representative of current and future conditions and that the training target is not too far from the truth.

One possible solution to obtain representative training data is to generate the training data set from simulated measurements. This simulation-based approach is followed with the NLIS (Non Linear Inference Scheme) algorithm developed by Crevoisier



(2023) for the retrieval of mid tropospheric $CO_2$ and $CH_4$ columns from IASI (Infrared Atmospheric Sounding Interferometer) and AIRS (Atmospheric InfraRed Sounder) measurements in the thermal infrared spectral region and by Xie et al. (2024) retrieving XCO2 from OCO-2 measurements over east Asia. However, building the training data set from simulations has not
only advantages. For the reasons discussed above, there are usually differences between simulated and measured spectra that cannot be explained by instrument noise. As with full-physics methods, these can affect the quality of the data products and again may require empirical bias correction.

A different approach was taken by David et al. (2021), who trained an MLP to retrieve XCO2 using actual measured OCO-2 data. This measurement-based learning has the potential advantage of virtually eliminating many of the sources of systematic
errors discussed above. However, it turned out that their ANN also appeared to have learned from spurious correlations, as it was unable to detect known local increases. After modifying the ANN and its input, Bréon et al. (2022) were able to show that their ANN was now able to detect local enhancements that were not part of the training data set. However, the authors also discuss that their ANN is not suitable for analyzing future data due to increasing $CO_2$ concentrations. In addition, they emphasize that despite the promising results, it is difficult to ensure that their ANN did not learn from a spurious correlation
again, especially since the reasons for the previous failure could not be fully determined.

Here we present the NRG-CO2M (Neural networks for Remote sensing of Greenhouse gases from CO2M) algorithm, which allows the use of actual measured spectra for training, but they are modified to cover a much larger range of XCO2 and XCH4 values. This type of hybrid learning combines the advantages of simulation-based and measurement-based learning: the characteristics of the actual measured spectra, including potential instrument effects, are preserved, almost any meaningful $CO_2$
and $CH_4$ concentration can be trained, and the variability of the training truth is dominated by prescribed artificial variations which can suppress learning from spurious correlations.

Nevertheless, our method also requires estimates of the true atmospheric concentrations to provide a representative training data set. These could be obtained in the same way as for empirical bias corrections (e.g., Noël et al., 2022; Schneising et al., 2023; Kiel et al., 2019) or as for the training data sets of other measurement-based ML methods (Bréon et al., 2022). Since
CO2M will not be launched until 2026, our study is based on simulated measurements from an extensive observing system simulation experiment (OSSE), which is a refinement of the OSSE described by Noël et al. (2024). As we are dealing with simulations, the true concentrations are known and, like Noël et al. (2024), we assume that there are no systematic errors in the training truth. Obviously, such errors would have the potential to reduce the accuracy of the prediction, but a realistic estimate of the to be expected error patterns of the training truth is difficult and beyond the scope of this study. We do, however, allow
for stochastic deviations of the training data from the true concentrations.

Sect. 2 describes the data sets and methods used, including the OSSE, the hybrid learning method, the transformation of the input data using principal component analysis (PCA), the method to modify the spectra, and the setup and training of the MLPs to determine XCO2, XCH4, and the corresponding uncertainties. Sect. 3 presents the results of the study and Sect. 4 provides a summary and conclusions.



## 2 Data sets and methods

### 2.1 Observing system simulation experiment

A comprehensive observing system simulation experiment (OSSE) was performed as part of a EUMETSAT study to develop the FOCAL CO2M retrieval algorithm (Noël et al., 2024). It contains simulated CO2M radiance data for nadir mode measurements over land, generated with the SCIATRAN RT model (Rozanov et al., 2017), taking into account realistic meteorology, albedo/BRDF, solar-induced chlorophyll fluorescence (SIF), aerosols, clouds, $CO_2$ and $CH_4$. The data set includes two years (2015 and 2020) of simulated CO2M orbit data with reduced sampling, hereafter referred to as subset data, as well as a high resolution (HR) scene simulated with the full CO2M sampling. It is an updated and extended version of the data set of simulated CO2I measurements used by Noël et al. (2024), so it is only briefly described here. The data set includes two years of subset data instead of one, and in addition to the simulated CO2I measurements, it has been extended to also contain simulated measurements from the MAP and CLIM instruments. In addition, the spectral variation of the albedo within the instrument bands is now more realistic and no longer constant.

The exact instrument characteristics of CO2M were not fully defined at the time of our study, so we used the MRD as a guide. The simulated main instrument CO2I consists of four imaging spectrometers for the wavelength ranges 405 nm–490 nm (VIS, $NO_2$), 747 nm–773 nm (NIR, $O_2$), 1590 nm–1675 nm (SWIR-1, $CO_2$ and $CH_4$) and 1990 nm–2095 nm (SWIR-2, $CO_2$) having spectral resolutions of 0.6 nm, 0.12 nm, 0.3 nm and 0.35 nm, respectively. In this study, we use CO2I data from the entire NIR band (1930 spectral features), and from the same wavelength ranges as used by Noël et al. (2024) in the SWIR-1 band (1590 nm–1670 nm, 931 spectral features) and SWIR-2 band (1990 nm–2090 nm, 953 spectral features). Since the VIS band is essentially intended for the determination of $NO_2$ atmospheric columns, it has not been simulated here.

For the hypothetical MAP instrument, we assumed that it has seven broadband channels (MAP1–7) with center wavelengths of 410 nm, 443 nm, 490 nm, 555 nm, 670 nm, 760 nm, and 865 nm, within which it determines the Stokes parameters I, Q, and U for each CO2I ground pixel at 45 equidistantly distributed along-track observation angles. In reality, MAP will have a higher spatial resolution, which will be aggregated to the CO2I measurements, and its MAP6 channel will only measure intensity, but this is not taken into account in this study.

The simulated CLIM instrument has three broadband channels (CLIM1–3), the first two of which spectrally coincide with MAP5 and MAP6. The central wavelength of CLIM3 is at 1370 nm in a strong absorption band of water vapor which makes this channel suitable for the identification of cirrus clouds. In reality, CLIM will have a much higher spatial resolution than CO2I, but this is also not taken into account in this study, so only CLIM3 provides additional information here.

An example of the complete simulated radiance measurements of a CO2I sounding, including co-located MAP and CLIM radiance measurements, is shown in Fig. 1.

### 2.1.1 Subset data

CO2I will have ground pixels with a spatial resolution of approximately 2 km × 2 km and will have 110 ground pixels per scan line across-track and each orbit will comprise approximately 9200 daytime scan lines along-track. In order to create





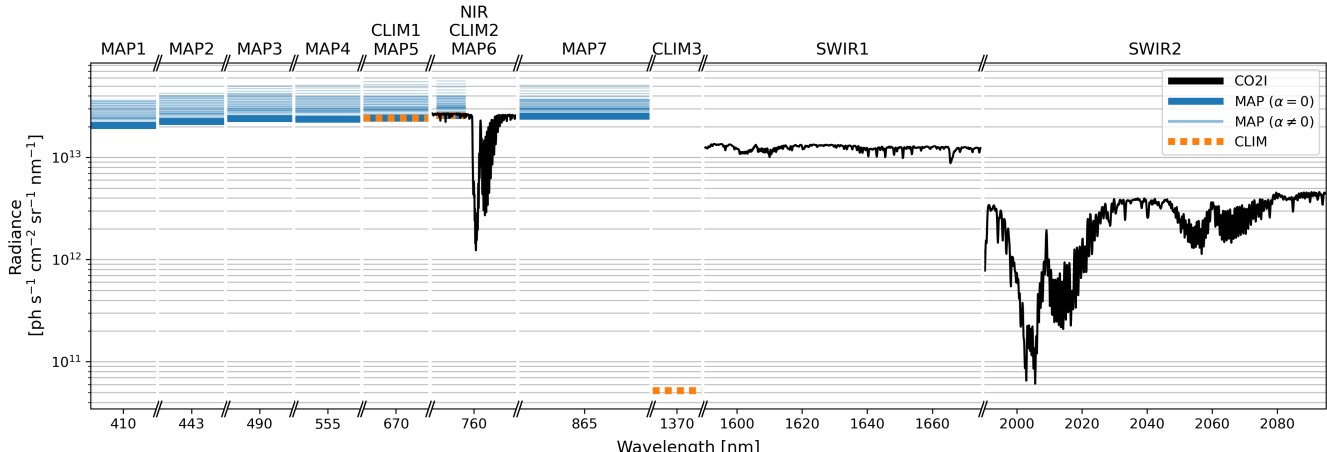

**Figure 1.** Simulated radiance measurements of a random CO2I sounding, including co-located MAP and CLIM radiance measurements. For MAP, the Stokes parameter I (total intensity) is shown for all simulated along-track observation angles $\alpha$.

representative training, test, and evaluation data sets, a minimum of two years of simulated CO2M data was desired. However, accurate RT simulations are computationally expensive, so it was not possible to simulate that many soundings in a reasonable

160 amount of time. Consequently, we adopted a strategy of subsetting the data set by simulating only every $15^{th}$ ground pixel across-track and every $20^{th}$ ground pixel along-track. This approach reduced the computational cost by a factor of 300 while largely maintaining the spatial and temporal coverage. For the SCIATRAN RT simulations, we used pressure, temperature, specific humidity, and cloud information from the ECMWF reanalysis ERA5 (Hersbach et al., 2020). Aerosol data were derived from CAMS' ECMWF atmospheric composition reanalysis EAC4 (Inness et al., 2019). $CO_2$ profiles were derived

165 from the CAMS global $CO_2$ atmospheric inversion v20r2 (Chevallier et al., 2005, 2010; Chevallier, 2013) and $CH_4$ profiles were obtained from the CAMS global $CH_4$ atmospheric inversion v20r1 (Segers, 2022). Surface reflectivity was modeled using the Moderate Resolution Imaging Spectroradiometer (MODIS) BRDF and albedo model parameter data set MCD43C1 version 6.1 (Schaaf and Wang, 2021). SIF was modeled using the MODIS Normalized Difference Vegetation Index (NDVI) MYD13C version 6.12 (Didan, 2021) as a proxy, following the approach outlined by RAL (2022). The resulting data set

170 includes approximately 2.13 million cloud-free soundings over land in 2015 and 2.15 million in 2020. In addition, the dataset also includes cloudy scenes that are sampled less densely depending on the cloud optical depth (COD), in order to emulate an imperfect cloud masking algorithm. Specifically, cloudy scenes are computed with a probability of $\mathrm{P_{COD}} = 1 - \mathrm{COD}$ but at least 0.05. This means that optically thin clouds are likely to make it into the data set, while the probability of an optically thick cloud is only 5%. This results in nearly half a million cloud contaminated land scenes per year.



### 2.1.2 Berlin high resolution scene

In addition to the subset data, we simulated a scene with the full CO2M sampling. It is a three-minute orbit granule with geophysical conditions of July $3^{rd}$, 2015, and since it includes Berlin (Germany), it is referred to as the Berlin HR scene. This scene is also used in EUMETSAT's CO2M preparation activities and HR model data are available for it. Our SCIATRAN input for this scene is the same as for the subset data, except for pressure, temperature, specific humidity, $CO_2$, and $CH_4$, which have been provided by EUMETSAT and which are based on the CAMS "nature run" model data with a spatial resolution of about 9 km (Agustí-Panareda et al., 2022). In particular, this means that the scene includes HR $CO_2$ and $CH_4$ signals, such as XCO2 plumes from power plants in Eastern Germany, which are not resolved in the subset data. For the Berlin HR scene we simulated 43671 soundings over land, of which 42398 are cloud-free.

## 2.2 Noise

For key parts of our study (e.g., PCA and ANN training), we need data scattering within realistic uncertainties, i.e., with noise distributions reflecting the expected statistical variability. In the case of the radiometric CO2M measurements, we used the same noise models as Noël et al. (2024) and Meijer et al. (2020). Based on the study by Salstein et al. (2008), we assume that the uncertainty of the dry-air column is 2.5‰. We further assume that the atmospheric temperature is uncertain by 1 K, realized by a shift in the entire profile. The atmospheric humidity is assumed to be uncertain by 10%, realized by profile scaling. For all observation angles, we define the uncertainty to be 0.1°. The target quantities XCO2 and XCH4 used as training truths are assumed to have uncertainties of 1 ppm and 5 ppb, respectively, which is somewhat larger than the differences between models and ground-based measurements found by Knapp et al. (2021) or Kulawik et al. (2016) and somewhat smaller than those found by Tu et al. (2020). The uncertainty of the $CO_2$ and $CH_4$ a priori profiles is accounted for by multivariate noise computed with the same a priori error covariance matrices used by Noël et al. (2024), scaled so that the a prior XCO2 and XCH4 scatter around the truth with a standard deviation of 4 ppm and 20 ppb, respectively. All uncertainty specifications in this section represent 1-sigma values of normally distributed random variables.

## 2.3 Modification of spectra

As discussed in Sect. 1, learning from simulated spectra can result in biases for the same reasons as for conventional full-physics retrieval methods namely because of inaccuracies in the RT and/or instrument simulation which is why we prefer to learn from measured spectra. However, this approach also has some potential disadvantages: XCO2 and XCH4 increase over time so that today's concentrations are not representative of the future; XCO2 and XCH4 may have correlations to quantities such as albedo or observation geometry from which an ANN can learn as efficiently as from spectral features; and uncertainties of the training truth may exist. For these reasons, we here use a method to modify measured spectra as if they include more or less of the target gases. Since CO2M was not yet operational at the time of our study, these measured spectra are simulations, namely the measurements of our OSSE simulated with SCIATRAN (Sect. 2.1).





An obvious way to modify a spectrum would be to use a synthetic Jacobian to simulate linear changes with respect to the geophysical state. However, due to nonlinearities of the RT, it is more promising to use the ratio of two synthetic spectra, i.e., a reference spectrum and a perturbed spectrum, for the modification. Since both the Jacobian and the ratio of the synthetic spectra depend on the geophysical state, it is necessary to first estimate it from the measurement.

In the following, we describe how we estimate the state from the measurement and generate the synthetic reference spectrum, how we perturb the state to make it representative of a wider range of conditions, how we generate the modified synthetic spectrum from it, and finally how we compute the modified measurement using the ratio of the synthetic spectra.

Let $I_m(x)$ be the measured CO2I intensity, i.e., a SCIATRAN simulated measurement of our OSSE (Sect. 2.1). It is a function of the true state $x$ including the true atmospheric concentration profiles of $CO_2$ and $CH_4$. In reality, when working with real
measurements instead of simulations, the true state is of course not known. We fit this measurement using the FOCAL retrieval as described by Noël et al. (2024) but with some adaptations guaranteeing that the vast majority of soundings are converging. Specifically, we enlarge the measurement error covariance by assuming an unrealistically large forward model uncertainty of 1% of the continuum radiance in all four fit windows and by allowing up to 40 iterations. The fitted radiance, i.e., the synthetic reference spectrum, is $I_f(\hat{x})$, where $\hat{x}$ is the retrieved state containing the retrieved concentration profiles $\hat{CO}_2$ and $\hat{CH}_4$. These
profiles consist of five layers, each representing the same number of dry-air particles.

It is important to note that FOCAL's RT is much simpler than the RT of SCIATRAN used to simulate the measurements so that a perfect spectral fit is usually not possible which is likewise the case when applying FOCAL to actually measured satellite data (Noël et al., 2021, 2022; Reuter et al., 2017a). As a result, the retrieved concentrations can significantly vary from the true atmospheric state, especially in scenes with enhanced scattering due to aerosols or clouds. This is much more the case
as it is for the studies of Noël et al. (2021, 2022, 2024) and Reuter et al. (2017a) because we here forced FOCAL to almost always converge and we applied no filtering or bias correction. However, this is not an issue for our study, since we are mainly interested in relative spectral changes, and we will show that it is even sufficient to use a simple non-scattering RT model that considers only gaseous absorption.

In the next step, we compute the perturbed concentration profiles $\tilde{CO}_2$ and $\tilde{CH}_4$ by adding delta profiles, which we calculate
as explained below using the example of $CO_2$.

- We randomly select two five-layer $CO_2$ profiles of the year 2015 from the SLIM (Simple cLImatological Model for atmospheric CO2 or CH4, Noël et al., 2022) climatological model and compute the difference concentration profile $\Delta CO_2$.

- We randomly increase or decrease $\Delta CO_2$ in the lowermost layer according to a normal distribution with a standard
deviation of 10 ppm, emulating the signal of a local source or sink.

- We compute the profile anomaly, i.e, we subtract the column-average of $\Delta CO2$ from $\Delta CO2$.

- We randomly shift the entire $\Delta CO_2$ profile according to a uniform distribution between -40 ppm and +40 ppm.



In this way, the shape of the delta profile $\Delta CO_2$ has large but not unrealistic variations with height and the variation of its column-average $\Delta XCO2$ is large enough to be representative for an atmospheric growth of many years.

$\Delta CH_4$ and $\Delta XCH4$ are calculated using the same method, but with all variations multiplied by a factor of $5 \times 10^{-3}$. This means that the standard deviation of the random $CH_4$ perturbation in the lowermost layer becomes $50\,ppb$ instead of $10\,ppm$, and the range of the uniform distribution for the random shift of the profile in the last step becomes [-200 ppb, +200 ppb] instead of [-40 ppm, +40 ppm]. Note that the perturbations of $CH_4$ are independent of those of $CO_2$.

As discussed above, the FOCAL-retrieved dry-air column-averages $X\hat{C}O2$ and $X\hat{C}H4$ of our study may be significantly

biased and we here consider them only as representative for the apparent light path. However, the corresponding climatological values $XCO2_{SLIM}$ and $XCH4_{SLIM}$ are relatively close to reality (Noël et al., 2022). Therefore, we scale the delta profiles $\Delta CO_2$ and $\Delta CH_4$ by a factor of $X\hat{C}O2/XCO2_{SLIM}$ and $X\hat{C}H4/XCH4_{SLIM}$, respectively, before performing the FOCAL forward run. This primarily effects scenarios with large deviations between retrieved and true concentrations.

$$C\tilde{O}_2 = C\hat{O}_2 + \Delta CO_2 \, \frac{X\hat{C}O2}{XCO2_{SLIM}} \tag{1}$$

$$C\tilde{H}_4 = C\hat{H}_4 + \Delta CH_4 \, \frac{X\hat{C}H4}{XCH4_{SLIM}} \tag{2}$$

These modified concentration profiles are part of the perturbed state $\tilde{x}$ which we use to perform an additional FOCAL forward run, i.e., RT and instrument simulation, in order to compute the modified synthetic spectrum $I_f(\tilde{x})$. This is then used to approximate what the measured radiance would look like if $\Delta CO_2$ and $\Delta CH_4$ were added to the true $CO_2$ and $CH_4$ profiles.

$$I_m(x + \Delta x) \approx I_m(x) \, \frac{I_f(\tilde{x})}{I_f(\hat{x})} \tag{3}$$

The quality of this approximation can be determined by comparing radiances approximated by Eq. 3 with corresponding SCIATRAN simulations. For this purpose, we selected one orbit of subset data of July $3^{rd}$, 2015 including many cloud-free scenes above Europe and Africa and shifted the entire $CO_2$ profile from -40 ppm to +40 ppm in steps of 1 ppm. Figure 2 shows an example spectrum of the approximation error in the SWIR-1 (a) and SWIR-2 (b) band for a 10 ppm shift of the $CO_2$ profile.

The figure shows that the approximation error, i.e., the difference between the approximation and the SCIATRAN simulation, is much smaller than the instrumental noise. As can be seen in panel c) and d) of that figure, the approximation error disappears for small profile shifts and steadily increases towards larger profile shifts. It is usually one order of magnitude larger in the SWIR-2 than in the SWIR-1. However, it is always significantly smaller than the instrumental noise. As an example, for a 10 ppm shift, the median $\chi^2$ amounts to 0.0003 in the SWIR-1 and 0.0041 in the SWIR-2. This means, the approximation is

valid within a range much larger than the current annual growth rate, thus allowing to generate a training data set from modified measured spectra being representative also for atmospheric conditions several years in the future.





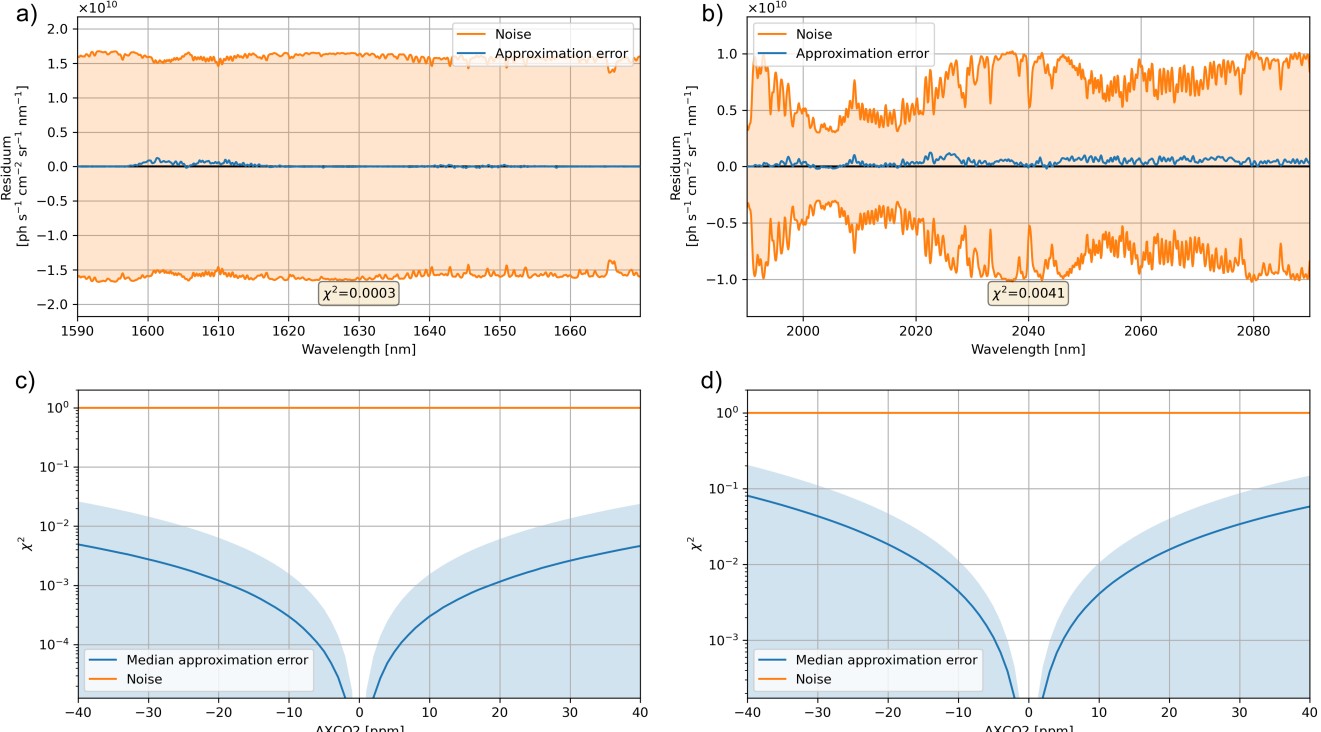

**Figure 2.** Spectrum of the approximation error (approximation minus SCIATRAN simulation) and instrumental noise for a typical scene and a 10 ppm shift of the $CO_2$ profile in the SWIR-1 (a) and SWIR-2 (b). Median (blue line) $\pm\ 1\sigma$ (light blue area) of the approximation error $\chi^2$ statistics for all cloud-free soundings of one orbit of subset data from July $3^{rd}$, 2015 as a function of the profile shift for the SWIR-1 (c) and SWIR-2 (d).

## 2.4 Principal component analysis

Atmospheric spectra, such as those measured by CO2I, contain a large amount of redundant information. In such cases, PCA is an efficient tool for dimensionality reduction without losing important information (e.g., Liu et al., 2006). It can significantly

reduce the size of the training data set and improve the learning efficiency of ANNs.

We used every seventh sounding of all even weeks in the 2015 subset data set and performed a PCA on various input data sets: the NIR band, the combination of both SWIR bands, the combination of all three bands, the MAP data, and the meteorological profiles of temperature and humidity.

The choice of the number of principal components used is not trivial and somewhat subjective. Using a large number of

components ensures that no information is lost, but the dimensionality reduction is small. If only a few components are used, the dimensionality reduction is high, but important information may be lost. We found that 25 components are sufficient for



**Table 1.** PCA results for various input data sets: NIR band, combination of both SWIR bands, combination of all three bands, MAP data, and meteorological profiles of temperature and humidity. Input data sets used for the baseline configuration of the ANNs (see later sections) are highlighted in gray. The table lists the number of components used, the corresponding unexplained variance and the $\chi^2$ of the reconstruction error, and the number of components for which the denoising error is minimal.

|  | NIR | SWIR1+2 | NIR+SWIR1+2 | MAP | Temperature | Specific humidity |
|---|---|---|---|---|---|---|
| Components used | 25 | 90 | 100 | 100 | 5 | 5 |
| Unexplained variance | $2.9\times10^{-9}$ | $3.5\times10^{-9}$ | $4.2\times10^{-9}$ | $2.4\times10^{-5}$ | $1.3\times10^{-2}$ | $8.8\times10^{-3}$ |
| Reconstruction error $\chi^2$ | $2.4\times10^{-4}$ | $9.9\times10^{-4}$ | $1.1\times10^{-3}$ | $9.5\times10^{-2}$ | n.a. | n.a. |
| Components min. denoising error | 17 | 55 | 75 | 120 | n.a. | n.a. |

the NIR band, 90 for the combined SWIR bands, 100 for the combination of all three bands, 100 for the MAP data, and 5 for the temperature and humidity profiles.

We based our choice of the number of components on calculations of the unexplained variance, the $\chi^2$ of the reconstruction
error, and the number of components that lead to a minimization of the denoising error. The unexplained variance is equal to one minus the explained variance which is commonly used in the context of PCA (e.g., Jolliffe and Cadima, 2016). The $\chi^2$ of the reconstruction error is calculated from the residual of the reconstructed and original measurements relative to the noise estimate of the measurements. The denoising error analyzes the residual between the reconstructed noisy data and the noise-free original data. It depends on the number of components used and reaches a minimum when the use of additional
components would predominantly lead to fitting noise but not signal (Aires et al., 2002; Di Noia et al., 2015). The $\chi^2$ of the reconstruction error and the denoising error were only determined for the radiation measurements where the noise estimates are known and reliable.

As an example, for the combination of the NIR and both SWIR bands, we find that when using 100 components, the fraction of unexplained variance amounts $4.2\times10^{-9}$. The $\chi^2$ of the reconstruction error is $1.1\times10^{-3}$ which means that the instrumental
noise can be expected to be about 1000 times larger than the reconstruction error. The denoising error becomes minimal when using 75 components. Selecting a significantly larger number of components can results in fitting noise, while selecting a significantly smaller number can result in loss of information. The results of all PCA studies are summarized in Table 1.

## 2.5 Artificial neural networks

### 2.5.1 Setup

In our study, we examined four different input compositions. The *baseline* setup is the standard setup used in this study. All other setups differ only in details in order to study their influence on the ANN's prediction quality separately. The *baseline* setup exists in a variant for XCO2 and a variant for XCH4. For simplicity, all other input setups exist only for XCO2.

The *baseline* input consists of the scores of the 100 most significant principal components (PCs) of the combined NIR, SWIR1, and SWIR2 spectra, the scores of the 100 most significant PCs of the MAP data, the CLIM3 radiance, the scores of





the five most significant PCs of the meteorological temperature and humidity profiles, the number of dry-air particles in the atmospheric column, the solar zenith angle, the satellite zenith angle, and the azimuth difference. As for conventional retrievals based on optimal estimation, the input also contains a noisy/uncertain a priori $CO_2$ or $CH_4$ profile (Sect. 2.2), which in our case consists of five atmospheric layers, each containing the same number of dry-air particles.

The *no MAP* input differs from the *baseline* input only in that it does not contain MAP and CLIM data. In addition to the missing MAP data, the *no NIR* input is also missing data from the NIR band. The *non scat.* setup is the same as the *baseline* setup, except that the modified spectra used for the training data set were generated by a FOCAL variant that only considers absorption but not scattering in the atmosphere.

All results were generated using MLP regressors with three hidden layers of 150, 30, and 150 neurons. The idea behind this ANN architecture is to improve the generalization capabilities of the network by adding a so-called information bottleneck in the middle layer, which holds the information of intermediate meta-parameters. Conceptually, there are parallels to first performing a conventional retrieval and then using the set of output parameters as input to a bias correction. We used the logistic, i.e., sigmoid, activation function and trained the MLPs with the Adam stochastic optimization method (Kingma and Ba, 2014) of the scikit-learn Python library (Pedregosa et al., 2011).

### 2.5.2 Training and test data set

To construct a representative and realistic training data set, we use noisy input and target data (see Sect. 2.2) which we construct from the data of all even weeks of the 2015 subset data set (Sect. 2.1.1). The data of the odd weeks are mainly reserved for testing. Separating the data sets on a weekly basis ensures that seasonal variations are sampled finely enough, while avoiding strong correlations between the two data sets that could occur with random sampling. The subset data contains a small fraction of cloudy scenes (Sect. 2.1.1), which we expect to be the case also in real data due to imperfect cloud masking. In order to create a realistic data set and to make the prediction less sensitive to residual cloud contamination, we filter out only clouds with an optical thickness greater than 0.05. From each remaining sounding, we generate ten soundings whose SWIR spectra have been modified as described in Sect. 2.3. Only these modified soundings, which have artificially increased XGAS variabilities, are the basis of our training data set.

### 2.5.3 Prediction of uncertainties

Interpretation of XCO2 or XCH4 satellite data requires appropriate uncertainty estimates. There are a number of ways to estimate the uncertainty of an ANN's prediction from the uncertainty of its input. The simplest approach is to present an existing ANN multiple realizations of the input, modified according to its error characteristics, and then statistically analyze the predictions. However, there are more sophisticated methods, such as the use of probabilistic ANNs (Mohebali et al., 2020). Here, we use a simple but efficient method by training MLPs to predict the XGAS uncertainties $\sigma$XCO2 and $\sigma$XCH4 from the same inputs used to predict XCO2 and XCH4, except that no a priori information is used. More specifically, we apply the XGAS MLPs to the test data set (Sect. 2.5.2) and compute the squared prediction mismatches (prediction minus training truth) $\Delta$XGAS$^2$ and use them as training targets for additional MLPs that predict the XGAS variances $\sigma$XGAS$^2$ as suggested by



Bishop (1996). The rationale behind this is that the expected value of $\Delta$XGAS is small, so the variance VAR(XGAS) can be approximated by the expected value of $\Delta$XGAS$^2$. We use data from the test period instead of the training period because the
prediction mismatches $\Delta$XGAS can be considered more realistic.

### 2.5.4 Column averaging kernel

In addition to reliable uncertainty estimates, the interpretation of XCO2 or XCH4 satellite data also requires information on the column averaging kernel (AK). The AK quantifies the retrieval's sensitivity to changes in the target gas concentration profile and is defined by

$$AK_i = \frac{1}{w_i} \frac{\partial X\hat{G}AS}{\partial GAS_i^t} \ , \tag{4}$$

where $X\hat{G}AS$ is the retrieved XGAS, $GAS_i^t$ the true gas concentration in height layer $i$, and $w_i$ the relative dry-air weight of that layer, i.e., the number of dry-air particles in sub-column $i$ divided by the total number of dry-air particles in the atmospheric column. In the context of retrieval comparison studies or surface flux inversions (e.g., Reuter et al., 2011; Bergamaschi et al., 2007; Wunch et al., 2011), $(1 - AK)$ can be interpreted as the influence of the a priori used. While we do not have direct
access to the column averaging kernel, the influence of the a priori $\partial X\hat{G}AS/\partial GAS_i^a$ can be easily determined numerically by predicting XGAS for perturbed a priori profiles GAS$^a$ and approximating:

$$AK_i \approx 1 - \frac{1}{w_i} \frac{\partial X\hat{G}AS}{\partial GAS_i^a} \ . \tag{5}$$

### 2.5.5 Postprocessing

As with conventional greenhouse gas retrieval algorithms, we filter out the least promising scenes during postprocessing. To
do this, we analyzed the 2015 evaluation data set (Sect. 2.5.6) and computed a threshold for the maximum allowed predicted uncertainty (Sect. 2.5.3) that filters out 10% of the cloud-free 2015 evaluation data. From the remaining data, we computed a threshold for the maximum allowed dependence on the a priori, which filters out another 11.11%. In this way, the most promising 80% of all cloud-free soundings remain after both filters. The thresholds are setup-specific and are listed in Table 2.

Additionally, we used the 2015 evaluation data set to compute a setup-specific overall offset (Table 2), which we subtract
from the prediction during postprocessing.

For each sounding, the a priori dependence is computed from the profile average sensitivity of the prediction to the a priori (Sect. 2.5.4). For example, if the dependence on the a priori was 5%, then adding 1 ppm to the CO$_2$ a priori would increase the XCO2 prediction by 0.05 ppm.

Similar to the dependence on the a priori, we compute the relative dependence of the prediction on the dry column. This
quantity specifies how dry column errors propagate to XGAS prediction errors. For example, if the dependence on the dry column was 5%, then a 1% error in the dry column would result in a 0.05% error in the predicted XGAS. This quantity is not used directly during post processing, but is analyzed when interpreting the results.





### 2.5.6 Evaluation data sets

We quantified the ANNs' prediction quality by applying them to three evaluation data sets that were not used for training: i) the unmodified 2015 subset data set (Sect. 2.1.1) which we divided into training and test period because it served as the basis for computing the training and test data sets (Sect. 2.3), ii) the unmodified 2020 subset data set (Sect. 2.1.1) with geophysical conditions and greenhouse gas concentrations not seen during the training, and iii) the Berlin HR scene (Sect. 2.1.2), also with geophysical conditions and greenhouse gas plumes that were not part of the training data set.

## 3 Results

For the input setups described in Sect. 2.5.1, MLPs with the properties described in the same section were trained to predict XCO2 and the associated uncertainty. In the case of the *baseline* setup, MLPs were also trained to predict XCH4 and its uncertainty. In order to analyze the prediction quality, the MLPs were applied to the evaluation data described in Sect. 2.5.6 and the prediction was compared with the truth.

Since the CO2M mission requirements are defined for cloud-free conditions, we filtered the evaluation data accordingly. Additionally, we applied the postprocessing filters described in Sect. 2.5.5. Most of the analyses were performed with noise-free input data, so the prediction errors can be considered as purely systematic.

The results for the 2020 subset evaluation data and the Berlin HR scene are the most conclusive because their input is the most independent of the training data set. In the following, we focus on the results for these data sets obtained with the *baseline* setup. However, Table 2 summarizes the main results for the analysis of all evaluation data sets and input configurations.

### 3.1 Column averaging kernels

We analyzed the XCO2 and XCH4 AKs of the 2020 subset evaluation data set. Figure 3a shows that the XCO2 AKs are close to optimal, i.e. close to unity, in large parts of the atmosphere. Significantly lower values are observed only in the stratosphere. The XCH4 AKs also decrease in the stratosphere, but show a slight overestimation of departures from the a priori in other layers (Fig. 3b).

### 3.2 Stochastic errors

In order to determine the overall retrieval precision due to instrumental noise, we predicted XCO2 and XCH4 from input with and without instrumental noise and calculated the standard deviation of the difference. For the postprocessed 2020 evaluation data set and the *baseline* setup, it amounts to 0.41 ppm for XCO2 and 2.74 ppb for XCH4.

As can be seen in Table 2, these values are basically identical to those obtained for the training and test periods of the 2015 evaluation data set and similar to those obtained for the Berlin HR scene.

The stochastic XCO2 error does not change for the *non scat.* setup (Sect. 2.5.1), but increases slightly to 0.45 ppm when the MAP instrument is not used. We see a more significant increase to 0.66 ppm when also not using the NIR band.



**Table 2.** Algorithm setup, postprocessing parameters, and main results generated from the evaluation subset data sets of 2015 and 2020 and from the Berlin HR scene for the *baseline* (gray), *no MAP*, *no NIR*, *non scat.* configuration.

|  | Baseline | | No MAP | No NIR | Non scat. |
|---|---|---|---|---|---|
| **Setup** | | | | | |
| Target | XCO2 | XCH4 | XCO2 | XCO2 | XCO2 |
| NIR | yes | yes | yes | no | yes |
| SWIR1+2 | yes | yes | yes | yes | yes |
| MAP+CLIM | yes | yes | no | no | yes |
| Modification method | scat | scat | scat | scat | non scat |
| **Postprocessing** | | | | | |
| Max $\sigma$XCO2\|$\sigma$XCH4 [ppm\|ppb] | 0.71 | 5.15 | 0.71 | 0.74 | 0.69 |
| Max a priori dependence [%] | 16.1 | 0.4 | 17.0 | 32.2 | 15.6 |
| Subtracted offset [ppm\|ppb] | -0.11 | 0.64 | 0.17 | 0.04 | -0.00 |
| **Evaluation results 2015 subset data** | | | | | |
| Soundings [#] | 1704695 | 1699842 | 1704595 | 1704181 | 1702525 |
| Throughput [%] | 80 | 80 | 80 | 80 | 80 |
| Precision train/test [ppm\|ppb] | 0.41/0.41 | 2.72/2.72 | 0.46/0.46 | 0.65/0.65 | 0.41/0.41 |
| Accuracy train/test [ppm\|ppb] | 0.39/0.42 | 2.20/2.37 | 0.42/0.46 | 0.38/0.42 | 0.39/0.43 |
| Mean bias [ppm\|ppb] | -0.00 | 0.00 | 0.00 | -0.00 | -0.00 |
| Mean a priori dependence [%] | 9.2 | -4.7 | 9.2 | 13.5 | 9.1 |
| Mean dry column dependence [%] | -6.2 | -4.7 | -16.5 | -60.6 | -5.6 |
| **Evaluation results 2020 subset data** | | | | | |
| Soundings [#] | 1704349 | 1724657 | 1691721 | 1685554 | 1679922 |
| Throughput [%] | 79 | 80 | 79 | 78 | 78 |
| Precision [ppm\|ppb] | 0.41 | 2.74 | 0.45 | 0.66 | 0.41 |
| Accuracy [ppm\|ppb] | 0.44 | 2.45 | 0.48 | 0.44 | 0.44 |
| Mean bias [ppm\|ppb] | 0.04 | 0.20 | 0.02 | -0.04 | 0.00 |
| Mean a priori dependence [%] | 9.6 | -4.8 | 9.3 | 14.2 | 8.7 |
| Mean dry column dependence [%] | -5.9 | -4.7 | -16.0 | -60.1 | -5.2 |
| **Evaluation results Berlin HR scene** | | | | | |
| Soundings [#] | 41757 | 41685 | 41390 | 41189 | 41888 |
| Throughput [%] | 98 | 98 | 98 | 97 | 99 |
| Precision [ppm\|ppb] | 0.44 | 3.12 | 0.47 | 0.70 | 0.43 |
| Accuracy [ppm\|ppb] | 0.31 | 1.72 | 0.40 | 0.39 | 0.34 |
| Mean bias [ppm\|ppb] | -0.18 | -2.13 | -0.29 | 0.22 | -0.36 |
| Mean a priori dependency [%] | 11.4 | -5.2 | 12.3 | 18.7 | 11.0 |
| Mean dry column dependency [%] | -3.9 | 0.8 | -17.3 | -70.8 | -2.4 |





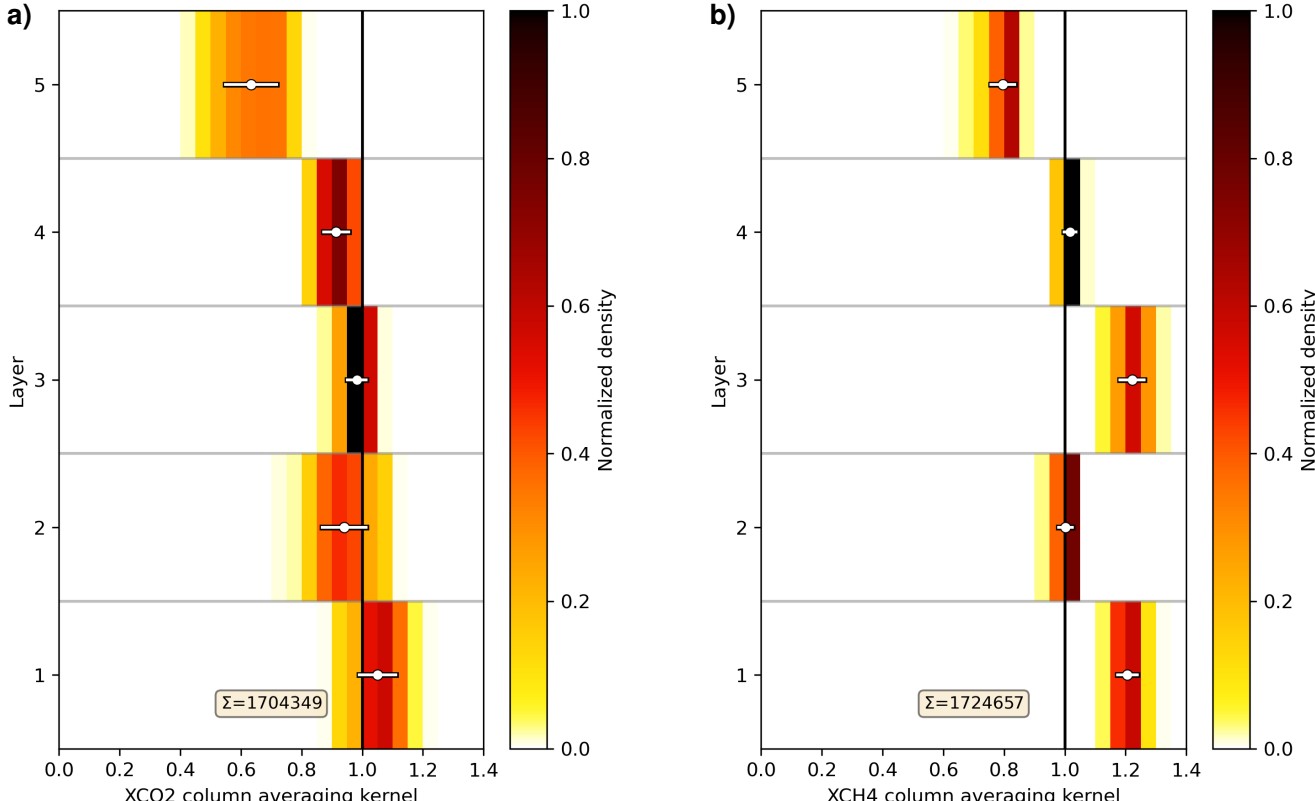

**Figure 3.** Normalized density distribution of the XCO2 (a) and XCH4 (b) column averaging kernels of all postprocessed soundings of the 2020 subset evaluation data set. Mean values and standard deviations are overlayed. The profile ing splits the atmospheric column in five layers, each containing the same number of dry-air particles. Layer 1 is the closest to the surface and includes the boundary layer, the stratosphere extends into layer 5. $\Sigma$ represents the total number of soundings.

In addition to the analysis of the overall precision, we validated the MLPs predicting the retrieval uncertainty of the individual soundings (Sect. 2.5.3). For this purpose, we defined 15 bins, each containing the same number of soundings, for the predicted
XCO2 or XCH4 uncertainty, respectively. For each bin, we determined the average predicted uncertainty, which we compared to the actual precision in that bin.

Figure 4 shows that the XCO2 and XCH4 uncertainties are well predicted by the MLPs. The predicted XCH4 uncertainties are almost on the spot. The predicted XCO2 uncertainties behave similarly but with a small offset of about 0.03 ppm.




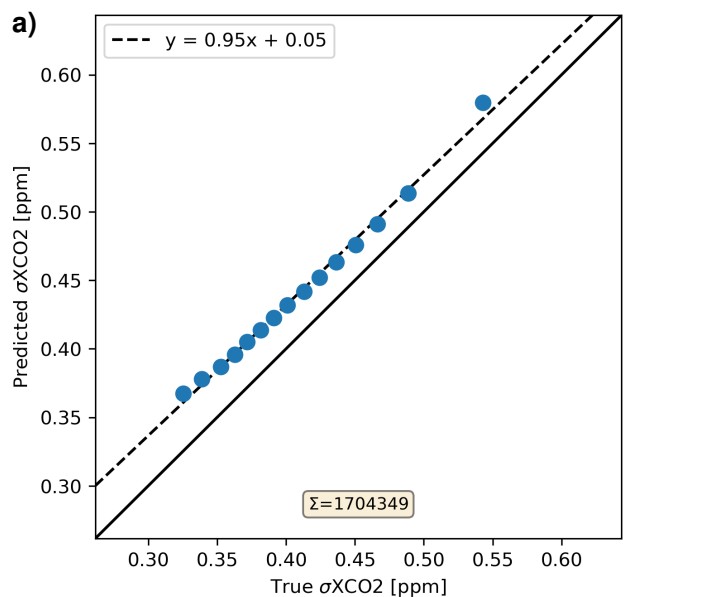
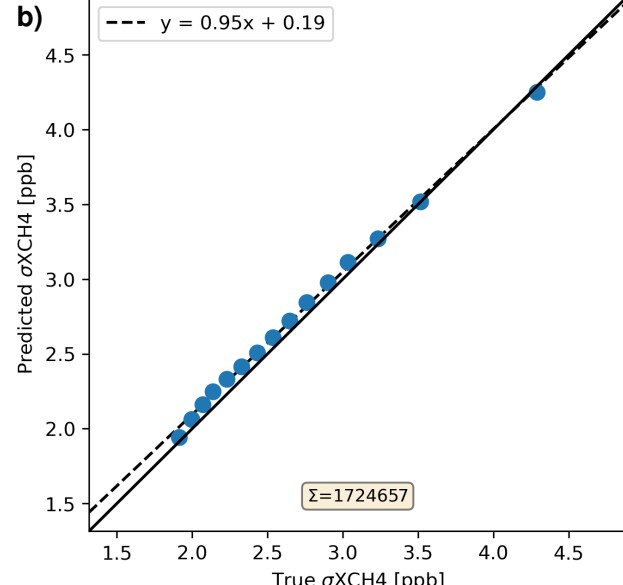

**Figure 4.** Comparison of the predicted and actual XCO2 (a) and XCH4 (b) retrieval uncertainties due to instrumental noise for the postprocessed 2020 subset data. $\Sigma$ represents the total number of soundings.

## 3.3 Systematic errors

### 3.3.1 Overall statistics

We compute systematic errors by comparing postprocessed predicted XCO2 or XCH4 values with corresponding true values for noise-free input data. Figure. 5 shows such a comparison for the 2020 subset data and the *baseline* setup.

With 0.04 ppm for XCO2 and 0.20 ppb for XCH4, the mean bias (prediction minus truth) for the 2020 subset data is negligible. It is no surprise, that this is also the case for the 2015 subset data, because this data set has been used to derive the postprocessing offset correction (Sect. 2.5.5). The mean bias for the Berlin HR scene is -0.18 ppm for XCO2 and -2.13 ppb for XCH4 (*baseline* setup).

Surface flux inverse modeling and emission estimation results are much more sensitive to spatially and/or temporally varying biases than to constant offsets. Therefore, we consider the standard deviation of the difference between predicted and true values as measure for the accuracy. For the 2020 subset data and the *baseline* setup it amounts to 0.44 ppm and 2.45 ppb for XCO2 and XCH4, respectively.

The accuracy values determined from the 2015 subset data are slightly smaller. The modification of the spectra used for the training can introduce small spectral errors (Sect. 2.3). These can erode the prediction quality the further we depart from





the concentrations of the training year 2015. Additionally, we observe that the prediction accuracy is about 10% better for the training than for the test period.

For the Berlin HR scene and the *baseline* setup, we obtain an accuracy of 0.28 ppm and 1.49 ppb for XCO2 and XCH4, respectively.

As can be seen in Table 2, the XCO2 accuracy depends only little on the setup, particularly for the subset evaluation data. At the first glance, this appears a bit surprising, because it would imply that the NIR band and the MAP instrument have only little influence on the systematic errors, which is not necessarily the case. Our analyses of systematic errors does not consider

systematic errors of the input such as the dry column or the a priori, which will exist in reality. When removing MAP from the input, the average dependence of the XCO2 prediction on the dry column increases from -5.9% to -16.0%. Additionally removing the NIR band further increases the dry column dependence to -60.1% and also increases the mean a priori dependence from 9.6% to 14.2%.

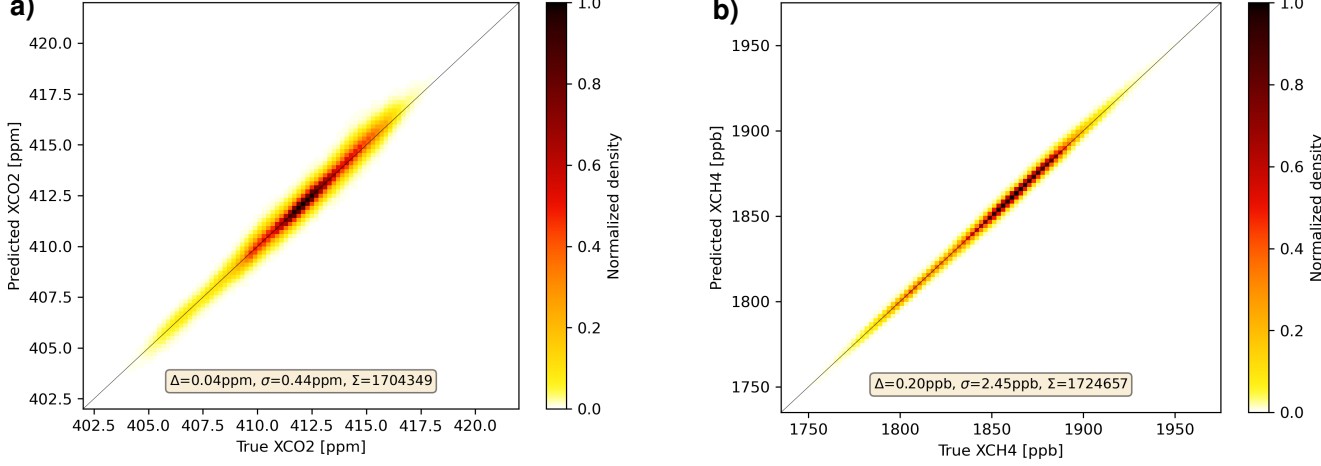

**Figure 5.** Comparison of postprocessed predicted XCO2 (a) and XCH4 (b) with corresponding true values for noise-free 2020 subset input data. $\Delta$ represents the average prediction error (prediction minus true), $\sigma$ the standard deviation of the prediction error, and $\Sigma$ the total number of soundings.

### 3.3.2 Large scale features

In order to investigate the spatial structures of the systematic errors we generated global maps for XCO2 (Fig. 6) and XCH4 (Fig. 7) showing postprocessed predicted and corresponding true values as well as their difference for noise-free subset input data of April and August 2020. First of all, the maps show a dense sampling because the postprocessing filters are designed to have a high throughput of about 80% for all cloud-free soundings (Sect. 2.5.5).

The maps of the predicted and true XGAS show the expected large scale features like low XCO2 values in northern mid

and high latitudes in August at the end of the growing season or relatively high XCH4 values in the tropics. The differences




between predicted and true XGAS values are generally much smaller than the large scale features. However, the differences are not purely random and exhibit some country to continental scale systematic features such as the small XCO2 and XCH4 high bias in Greenland in April or the small XCH4 high bias in northern Africa in August.

There are some similarities between the XCO2 and XCH4 bias patterns which may indicate, that some systematic errors could cancel out in a proxy product when using, e.g., the ratio XCH4/XCO2 as training target.

The global monthly average biases are small and the corresponding standard deviations similar to the annual averages.

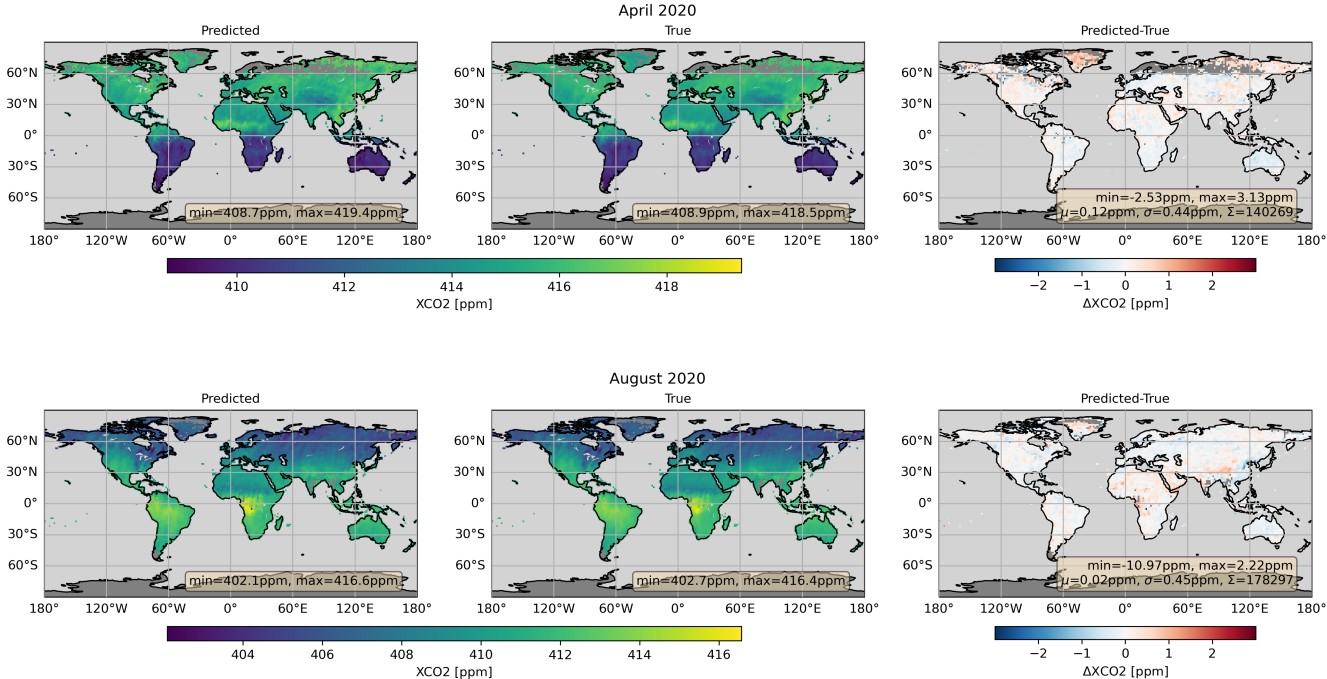

**Figure 6.** Global maps of postprocessed predicted XCO2 (left) and corresponding true values (middle) as well as their difference (right) for noise-free subset input data of April (top) and August (bottom) 2020. $\mu$ represents the average prediction error (prediction minus true), $\sigma$ the standard deviation of the prediction error, and $\Sigma$ the total number of soundings.

### 3.3.3 Seasonal cycle

Systematic errors may also have a seasonal component, e.g., due to seasonal variations in illumination conditions, albedo, or aerosols. Figure 8 shows the XGAS prediction error as a function of the week in the year 2020. According to this figure, the average systematic XCO2 prediction error slowly drifts around zero, with largest values of about 0.2 ppm in late (northern hemispheric) spring and smallest values of about -0.1 ppm in autumn. The standard deviation of the XCO2 error is larger in spring and summer (up to about 0.55 ppm) compared to autumn and winter (down to about 0.40 ppm). Various reasons can cause this behavior, e.g., sampling in summer covers a wider latitude range and, therefore, also more surface types and





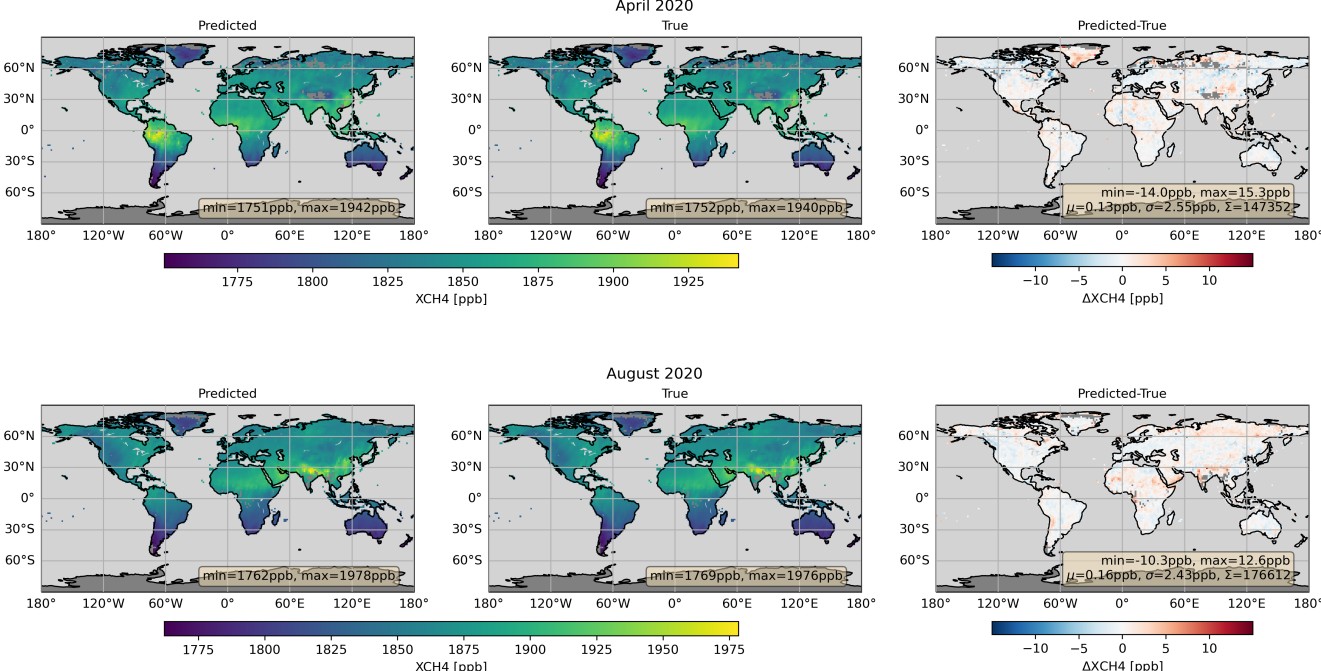

**Figure 7.** Global maps of postprocessed predicted XCH4 (left) and corresponding true values (middle) as well as their difference (right) for noise-free subset input data of April (top) and August (bottom) 2020. $\mu$ represents the average prediction error (prediction minus true), $\sigma$ the standard deviation of the prediction error, and $\Sigma$ the total number of soundings.

observation angles than in winter, the $CO_2$ profiles vary more during the (northern hemispheric) growing season. The weekly
average prediction error of XCH4 has no clear seasonal cycle and is always smaller than $\pm 1$ ppb. Its standard deviation varies between about 2.2 ppb and 3.3 ppb.

### 3.3.4 Aerosols

Aerosols modify the light path and can for this reason be an important source of XGAS retrieval errors. Figure 9 shows the XGAS prediction error as a function of aerosol optical depth (AOD) for noise-free 2020 subset data. As can be seen, the XCO2
average prediction error stays close to zero up to an AOD of 0.2. For larger AODs up to 0.5, the average prediction error steadily increases to values of about 0.1 ppm. The standard deviation of the prediction error increases with AOD from about 0.35 ppm to about 0.60 ppm. The average XCH4 prediction error is usually below $\pm 0.5$ ppb and its standard deviation increases from about 2.0 ppb for basically aerosol-free scenarios to about 3.2 ppb for scenarios with an AOD of up to 0.5.



**Figure 8.** XCO2 (top) and XCH4 (bottom) prediction error as function of the week for noise-free 2020 subset input data. Dots and bars represent mean and standard deviation. $\Sigma$ represents the total number of soundings.





**Figure 9.** XCO2 (top) and XCH4 (bottom) prediction error as function of AOD for noise-free 2020 subset input data. Dots and bars represent mean and standard deviation. Σ represents the total number of soundings.





### 3.3.5 Berlin HR scene

Since the AKs are close to unity in large parts of the atmosphere (Fig. 3), the prediction can be considered as dominated by the measurement, but not the a priori. In order to illustrate this, we used scene-wide constant a priori profiles instead of the true concentration profiles to analyze the Berlin HR scene. Specifically we used the scene-wide average true $CO_2$ and $CH_4$ concentration profiles as a priori.

Figures 10 and 11 show that the predictions reproduce the true concentrations well, even though the meteorological condi-
tions and gas concentrations, including plumes from strong $CO_2$ and $CH_4$ sources, were not part of the training data or the a priori.

The variability of the difference structures are much smaller than the variability of the atmospheric signals. The XCO2 prediction error has a standard deviation of 0.31 ppm and an average of -0.18 ppm. It shows no obvious correlations with the XCO2 pattern, especially the $CO_2$ plumes from the coal-fired power plants Jänschwalde, Schwarze Pumpe, and Boxberg in
eastern Germany.

The prediction error of XCH4 is on average -2.13 ppb and has a standard deviation of 1.72 ppb. However, the map of the XCH4 prediction error (Fig. 11) shows an interesting feature at about 50.53°N, 13.61°E in the Czech Republic. There is a strong $CH_4$ plume at this position, the strength of which is obviously overestimated by the prediction.

As a reminder, the AKs describe the behavior of the retrieval to over- or underestimate differences between the true and the
a priori concentrations. The plume stands out from the scene average concentrations, i.e., the a priori, at roughly 90 ppb. The XCH4 AKs in the lowermost layer can have values of up to 1.3, which would result in an overestimation of the departure from the a priori by 30%, i.e., 27 ppb in this case. It shall be noted that this would not result in an overestimation of the emission strength, if AKs are considered appropriately.

When using the true $CO_2$ and $CH_4$ profiles as a prior, the difference maps look similar except that there is no such overesti-
mation of the $CH_4$ plume because the departure from the a priori gets much smaller (not shown).

### 4 Summary and Conclusion

In preparation for the analysis of the large amount of radiance measurements from the CO2M satellite mission, we developed the computationally efficient ANN-based algorithm NRG-CO2M to retrieve XCO2 and XCH4 with high accuracy and precision and high data yield.
It adapts a hybrid learning method that is designed to use measured spectra modified to represent a wider range of XCO2 and XCH4 values. The approach combines the advantages of simulation-based and measurement-based learning, preserving the characteristics of the real spectra, including instrumental effects, while allowing learning over a wide range of $CO_2$ and $CH_4$ concentrations.

It minimizes learning from spurious correlations by dominating the variability of the training data with prescribed artificial
variations. However, the method still requires estimates of the true atmospheric concentrations for a representative training data set, which can be obtained similarly to methods used for empirical bias corrections.



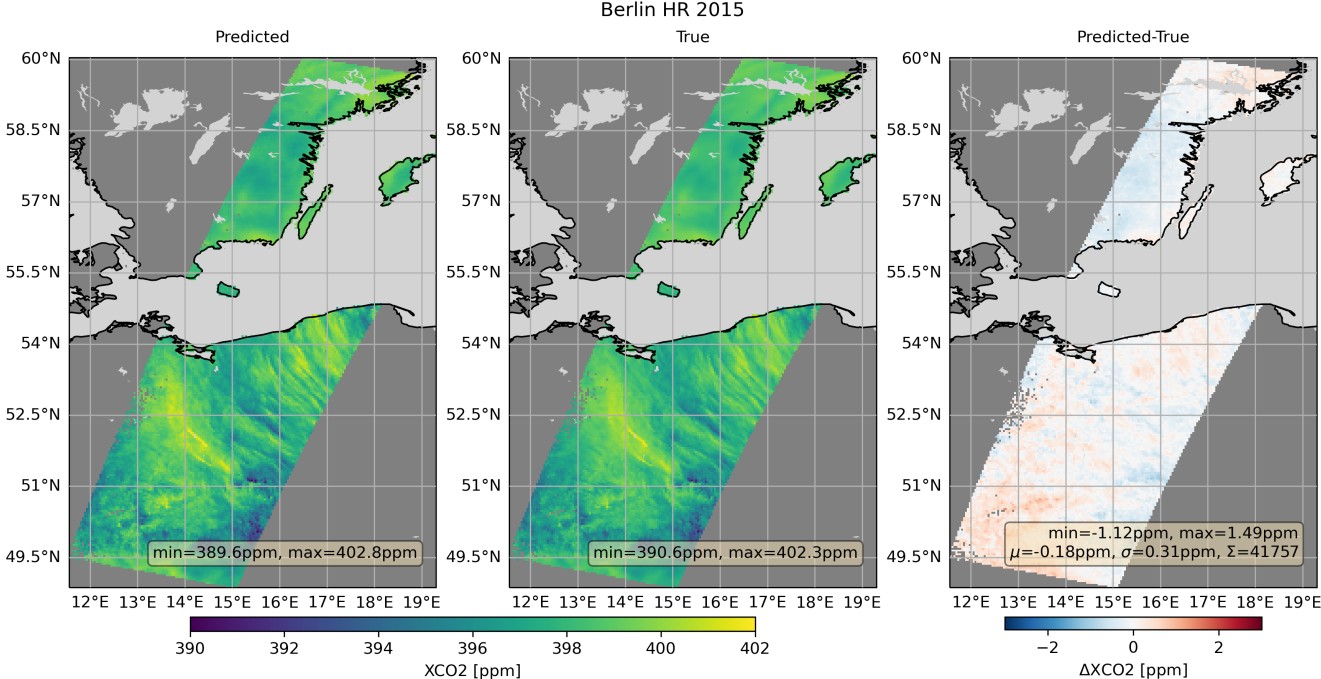

**Figure 10.** Postprocessed predicted XCO2 (left) and corresponding true values (middle) as well as their difference (right) for noise-free Berlin HR input data.

Since the CO2M mission will not be launched until 2026, the study is based on simulated measurements from an OSSE. These simulations assume no systematic errors in the training truth, although they do account for stochastic deviations from true concentrations.

Due to the design of the OSSE used, we have focused in this study only on soundings over land in nadir geometry but the methods presented should be equally applicable to measurements over water surfaces and under glint conditions.

From our analyses of the 2020 subset data, we find that the systematic XCO2 and XCH4 errors scatter with a standard deviation of 0.44 ppm and 2.45 ppb, respectively. This compares to mission requirements for spatio-temporal systematic errors below 0.5 ppm for XCO2 and 5 ppb for XCH4 (MRD, Meijer et al., 2020). The average single sounding precision is 0.41 ppm

for XCO2 and 2.74 ppb for XCH4, compared to mission requirements for stochastic errors due to instrumental noise of less than 0.7 ppm for XCO2 and 10 ppb for XCH4 defined for a specific vegetation scenario (MRD, Meijer et al., 2020). Therefore, we conclude that the proposed retrieval method has the potential to meet the demanding CO2M mission requirements for systematic and stochastic XCO2 and XCH4 errors.

Our results are qualitatively similar to those of Noël et al. (2024). They estimated the spatio-temporal systematic errors of

their FOCAL setup to be 0.5 ppm for XCO2 and 3.7 ppb for XCH4 and the stochastic errors to be 0.5 ppm for XCO2 and 5.0 ppb for XCH4. However, unlike Noël et al. (2024), we did not divide the systematic error into long correlation length parts,





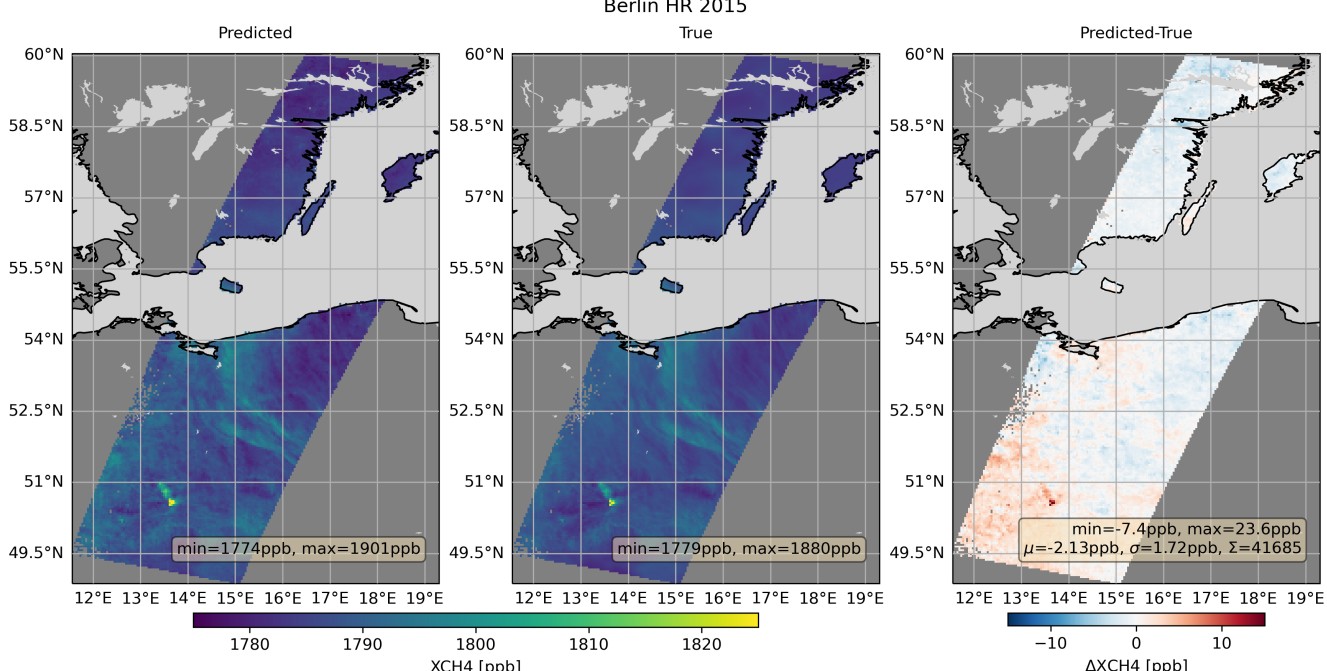

**Figure 11.** Postprocessed predicted XCH4 (left) and corresponding true values (middle) as well as their difference (right) for noise-free Berlin HR input data. $\mu$ represents the average prediction error (prediction minus true), $\sigma$ the standard deviation of the prediction error, and $\Sigma$ the total number of soundings.

which are relevant for the application of large scale surface flux inversions, short correlation length parts, which are relevant for the application of small scale (e.g., point source) emission estimation. Our results for the Berlin HR scene illustrate how this affects estimates of the relevant systematic errors. The total systematic error in this scene consists of a variable part scattering

with a standard deviation of 0.28 ppm for XCO2 and 1.49 ppb for XCH4 and a scene-wide bias of -0.18 ppm for XCO2 and -2.12 ppb for XCH4. However, only the variable part of the systematic error is relevant for the application of small scale (e.g., point source) emission estimation. It should also be noted that our postprocessing is designed to globally reject about 20% of the least promising soundings, compared to a rejection rate of about 37% used by Noël et al. (2024).

We trained the ANNs with (modified) spectra from the year 2015. Consequently, it can be expected that the modification

error becomes more important the further we deviate from the training year. Nevertheless, we observe that the quality of the prediction erodes only slowly, because compared to 2020, the accuracy is only slightly better during the test period in 2015 (0.02 ppm for XCO2, 0.08 ppb for XCH4) and the precision is the same. This shows that the introduced spectrum modification method is able to efficiently improve the representativeness of the training data for future concentrations years ahead.




We use a conventional XCO2 and XCH4 retrieval to modify the spectra used for the training data set. It is a variant of the
FOCAL algorithm described by Noël et al. (2024), which takes into account scattering in the atmosphere. However, our results
show that using an absorption-only retrieval for this task leads to results with essentially the same accuracy and precision.

As a test, we also trained ANNs without MAP data. This had an apparently small effect on accuracy and precision, which
is not consistent with the results of Lu et al. (2022), whose retrieval method became significantly less accurate under these
conditions. However, we observe that the dependence of the XCO2 prediction on the dry column increases considerably when
MAP is not used, which will introduce systematic errors in reality when perfect knowledge of the dry column cannot be
expected. Additionally removing the NIR band further increased the dependence on the dry column, but also the dependence
on the a priori, making it less likely to meet the CO2M mission requirements.

It is expected that several thousand CPU cores will be required to continuously analyze the data stream from a single
CO2M satellite using the conventional full-physics algorithms, which are currently being implemented by EUMETSAT. In
comparison, the computational requirements of the presented ANN retrievals, once trained, are negligible and can be considered
to be driven by pre- and postprocessing as well as input and output operations.

However, the development of neural networks for retrieval of greenhouse gases from satellite-based measurements of re-
flected solar radiation in the NIR and SWIR is still in its early stages, while there is much experience with full-physics meth-
ods. This includes how they respond to instrumental effects such as spectral artifacts or temporal changes, as well as machine
learning strategies for bias correction. In addition, it should be noted that the results of Noël et al. (2024) suggest that one
month of training data may be sufficient for a machine learning-based bias correction. In contrast, we used one year of training
data for our ANN-based approach, which can make a difference in the early phase of a satellite mission when little data is
available.

In summary, the proposed retrieval algorithm can be seen as a promising candidate to meet the high accuracy and precision
mission requirements while providing high data yield and negligible computational requirements, making it a valuable addition
to the ensemble of conventional algorithms.

*Data availability.* The data used in this study are available upon request from the corresponding author.

*Author contributions.* Experimental setup, data analysis, design and operation of the ANN retrieval, writing the paper: MR. Design and
creation of the OSSE data base, design and operation of the FOCAL retrieval: MR, SN, MH. Provision of the CO2M geolocation information
and CO2M instrument expertise: RL. Interpretation of the results, improving the paper: All authors.

*Competing interests.* The authors declare no competing interests.



*Acknowledgements.* This research was supported by the European Union Copernicus program through EUMETSAT contract no. EUM/-CO/19/4600002372/RL, and by the state and the University of Bremen. Parts of the calculations reported here were performed on HPC facilities of the IUP, University of Bremen, funded under DFG/FUGG grant nos. INST 144/379-1 and INST 144/493-1. We used meteorological data from the ECMWF ERA5 reanalysis, aerosol data from CAMS' ECMWF EAC4 atmospheric composition reanalysis, $CO_2$ concentrations from the CAMS global $CO_2$ atmospheric inversion v20r2, $CH_4$ concentrations from the CAMS global $CH_4$ atmospheric inversion v20r1, NASA's MODIS BRDF and albedo model parameter data set MCD43C1 v6.1, and the NASA NDVI data set MYD13C v6.12. Deepl (www.deepl.com) has been used for linguistic improvements.



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
