# Peer review of "Retrieving the atmospheric concentrations of carbon dioxide and methane from the European Copernicus CO2M satellite mission using artificial neural networks"

_EGUsphere, 2024_

## Referee Comment (RC1)

Review of "Retrieving the atmospheric concentrations of carbon dioxide and methane from the European Copernicus CO2M satellite mission using artificial neural networks"

This paper describes a neural-network (NN) based direct retrieval of column mean concentrations of carbon dioxide (XCO2) and methane (XCH4), directly from spectra from a simulated CO2M satellite, using all of its instruments: its spectrometer (CO2I), multi-angle polarimeter (MAP), and it's cloud instrument as well. It uses realistic OSSEs that include aersols (but not clouds, it seems) to form a challenging dataset, though the authors are clear they do not include instrument artifacts in their simulations. Their predicted XCO2 and XCH4 values are excellent, and exceed the performance requirements of CO2M. Additionally, they provide a method to give both posterior uncertainties and averging kernels for their predicted XCO2 and XCH4 values. Notably, they also present a novel method to extend their training dataset forward in time, in order to avoid the problem of needing to retrain every couple of years as greenhouse gas concentrations continue to increase.

This paper is well-written, and does not suffer from some of the problems of previous machine-learning based XCO2/XCH4 retrieval papers. They have described their methods well, and the conclusions are indeed promising. However, they do have an implicit conclusion that the MAP instrument doesn't seem to be required, at least under their assumptions, as the error statistics are similar with and without MAP, and their claim that the lesser dependence of the dry-air column with MAP doesn't stand up to further scrutiny. They need to more fully examine this implicit conclusion. After they have sufficiently addressed this and my additional comments below, I recommend the manuscript for publication, as it will be an important addition to the literature.

**All Comments**
- Abstract: It would be helpful to say quickly in one sentence how the OSSE is set-up to make life "difficult" for the retrieval (includes plumes, realistic aerosols, etc) to make the results more meaningful. Also, has any other retrieval method demonstrated they can meet the accuracy and precision requirements of CO2M, or is this the first? If it is the first, it's important to say so. Though it looks like RemoTAP also does, based on Lu et al 2022, is that also your read? If so then I guess say nothing…
- Abstract: I think it would be good to modify the abstract and conclusions to make it clear that you would have to re-train with real data once CO2M data are available, and that could change the storyline because of instrument artifacts, lack of sufficiently good training data (do you use TCCON, or a model, or…?). So while this is a solid proof-of-concept, we can only really believe the amazing results once you apply it to real data somehow.
- Abstract: The sentence "We employ a hybrid learning approach that combines advantages of simulation-based and measurement-based training data to ensure coverage of a wide range of XCO2 and XCH4 values making the training data also representative of future concentrations." Is important! But it downplays the excellent work you've done here. Even if your NN approach didn't work, this one thing is great

and could be utilized by any researcher trying to do direct ML-retrievals of GHGs. Maybe change to "We created a novel hybrid learning approach…". You could also add a sentence like "This method could easily be applied by future researchers training ML-based GHG retrievals, to avoid this common problem." Or something to that effect. I think it's just important to highlight this contribution to the literature, in addition to your actual ML model.

- Abstract: I think you should also add a sentence to the effect of "Our ML model also provides accurate estimates of both the noise-driven uncertainties and the averaging kernels of XCO2 and XCH4 for each sounding." This is an important aspect of your model; not all ML models do this.

- L43: BRDF → surface BRDF
- Fig1: For the love of god, please convince your CO2M colleagues to work in W m^-2 um^-1 sr^-1 units. We messed this up for OCO2/3. You can right this wrong.
- Page6: How are clouds modeled in the radiative transfer? Do they come from CAMS? From where does the effective radius for water and ice come? Clouds were excluded in Noel et al (2024) for the FOCAL tests. It seems like you are trying to include them here, so more details are welcome, since this is a specific difference to Noel et al.
- Section 2.2. It's not clear how these uncertainties in dry-air column, temperature, co2 profile etc are used. Are you saying that you stochastically apply these terms to the truth training data before you simulate the spectra? Or that you stochastically supply them as input to the NN predictions, so the NN doesn't have perfect knowledge of things like temperature profile, etc, when performing a retrieval on a given sounding? Please be clear. A flowchart might be helpful here. I think you ARE supplying these to the NN (you seem to say this in section 2.5) but please be explicit here. I think also saying WHY you need to supply this information is important.

  Side note: I worry that you are telling your NN technique the answer *by construction* for each sounding, by supplying "truth data + gaussian noise" to it. It might be fine. But your "truth data + gaussian noise" for temperature, co2, surface pressure, etc, is not biased; there are no systematic errors. Instead, I would prefer that you had used a completely different model for your "prior information". For instance, CarbonTracker for CO2, MERRA-2 for Temperature, humidity, surface pressure, etc. Your hypothesis would be that it doesn't matter, but to me, that isn't clear.

- Near line 360. Feel free to add a contextual comment like: "For comparative purposes, the dry air column dependence for the operational OCO-2 XCO2 retrieval (v11.1) is roughly 85%, making it highly dependent on the accuracy of the prior meteorology, the prior surface elevation, and the instrument pointing (Jacobs et al., 2024, https://amt.copernicus.org/articles/17/1375/2024/)."

- Near line 420. I don't get why removing the NIR band doesn't increase the dependence on the dry air column to 100%! Where is information on the dry column coming from?

I guess from the fact that your prior co2 profiles are pretty good, so it can partially deduce the dry column from the co2 bands alone?

Also, regarding the increase in the dry column dependence when you remove MAP, from 6% to 16%. Typical surface pressure uncertainties are on order 1-2 hPa (or often even smaller). +- 2 hPa is 2/1000 roughly, and 10% of this is 2/10000. For a typical XCO2 of 400 ppm, this would induce an uncertainty of 0.08 ppm. This implies that removing MAP from CO2M which add an additional +- 0.08 ppm uncertainty to XCO2, due to errors in the prior surface pressure, relative to the with-MAP case. Which basically means that, according to your analysis, MAP really is not necessary. That's a pretty big conclusion that you are currently glossing over. Please address this directly in the manuscript. Presumably its due to some assumption you've made?

FYI this also affects your interpretation in the conclusions (near 520), where you are implying that this is an important difference for the no-MAP case. It's really not, honestly. OCO-2/3 would kill to only have a 15% dependence on the dry air column, which leads to nearly negligible errors in the target gases.

- Near Line 470, and Figures 10+11. Can't you plot the AK-corrected Truth minus Prediction, instead of straight truth – prediction? You should! I *always* do this in my OSSE experiments, it is important. It would also show if your hypothesis is correct on the source of this hotspot in the difference plot of figure 11. In fact a comparison of these two plots (with and without AK-correction) would be very illuminating. Your statement on using the true profiles as prior comes close to accomplishing this, but is not nearly as powerful. Plus, you are expecting modelers to make the AK correction; therefore I think It's important to set a good example and do the same, and show the effect when you don't.

- L502: short correlation length parts → or short correlation length parts

- I think the conclusions section really needs a paragraph on what it would take to "operationalize" this algorithm for real satellite data. Presumably you would train it on observed spectra, along with your method to extend it to larger truth values of XCH4 and XCO2? What would you use for the training truth: TCCON, Models, something else? Would your methods to get at the AK and posterior Xgas uncertainties still work? Would you have any reason to expect worse performance?

---

## Referee Comment (RC2)

Review of EGUsphere-2024-2365

Retrieving the atmospheric concentrations of carbon dioxide and methane from the European Copernicus CO2M satellite mission using artificial neural networks

By Maximilian Reuter, Michael Hilker, Stefan Noël, Antonio Di Noia, Michael Weimer, Oliver Schneising, Michael Buchwitz, Heinrich Bovensmann, John P. Burrows, Hartmut Bösch, and Ruediger Lang

*General comments.*

The authors present a new retrieval method from Level 1B of CO2M data. I understand that this activity is important to retrieve a good $XCO_2$ and $XCH_4$ products from CO2M data with acceptable computational time. However, some description and assumption are unclear or missing in the text. For example, to retrieve the $XCO_2$ and $XCH_4$, the instrumental model is very important. In this manuscript, only the random noise is assessed. The authors should concern the other parameters at least the uncertainly of instrumental line shape function and its wavelength depended response. In addition, the authors were used the actual space-based observation data such as OCO-2 during the FOCAL development. To evaluate the new NRG-CO2M algorithm with actual space-based observation data with realistic uncertainty is also important and informative. However, the authors are only focused the simulation-based dataset. I understand the CO2M will not be launched until 2026. The authors should be considered the evaluation plan with the updated instrumental model data and the realistic characterization error, and these impact on the NRG-CO2M processing. Furthermore, the application for the actual space-based observation dataset, currently available dataset, is also informative and productive for the evaluation purpose. The authors should be considered the evaluation plan for the NRG-CO2M with currently available observation dataset. I recommend the authors will add the sentences and clarify for some of unclear sentences.

For these reasons, I recommend this paper for publication with minor changes to the technical content.

*Specific comments.*

*Abstract*

1. Page 1, line 13: Spell out first for "NRG-CO2M". -> Neural networks for Remote sensing of Greenhouse gases from CO2M (NRG-CO2M)

2. Page 1, line 19: The definition of "spatio-temporal systematic errors" is unclear. The authors should add the definition or more clear explanation for the condition.

*1.  Introduction*

3. Page 2, line 39: add the "," between "5ppb" and "respectively".

4. Page 2, line 41: Spell out first for "CO2I".

5. Page 2, line 42: Spell out first for "MAP".

6. Page 2, line 42: Spell out first for "BRDF".

7. Page 2, line 43: Spell out first for "CLIM".

8. Page 2, line 44: XCO2 or XCH4 -> XCO2 and/or XCH4

9. Page 2, line 47: 2017b,a -> 2017 a, b

10. Page 2, line 49: Spell out "EUMETSAT".

11. Page 2, line 57: The meaning of "3D effects" is unclear. The authors should add the explanation.

12. Page 3, line 61: Spell out first for "OCO-2".

13. Page 3, line 62: Spell out first for "GOSAT".

14. Page 3, line 77: the meaning of "meteorology and angles" are unclear. The authors should add the explanation.

15. Page 3, line 83: Krasnopolsky and Schiller (2003). -> (Krasnopolsky and Schiller, 2003).

16. Page 4, line 116: Spell out first for "OSSE".

17. Page 4, line 116: In the previous works, the authors were developed FOCAL full physics algorithm. During the development phase of FOCAL, the authors are actually used the space-based observation data such as OCO-2 and GOSAT. To evaluate the new NRG-CO2M algorithm with actual space-based observation data is quite realistic and import. However, the authors are only focused the

simulation-based dataset. So, the authors should be considered the evaluation plan with actual space-based observation dataset or current limitations.

**2. Data sets and methods**

18. Page 5, line 154: What is the instrumental line shape model? It also has several uncertainties. It is not clear how to take account spectrally depended uncertainties. The authors should add the explanation.

19. Page 7, line 193: How to consider the bias in a priori? Especially in the future prediction, not only a standard deviation but also the bias has to be considered. The authors should add the explanation.

**3. Results**

20. Page 18, Figure 5: How is the slope? It seems that the linearity can be directly estimated from this analysis. However, it is not mentioned in the text.

End of document

---

## Author Comment (AC1)

First of all, we thank Christopher O'Dell (reviewer 1) for his effort in carefully reviewing our manuscript and his constructive comments.

**Point-by-point answers to the comments of reviewer 1**

**General points**

**Authors:** All points raised in the general part correspond to specific comments and are, therefore, discussed in the next section.

**All comments**

***Reviewer 1:*** *Abstract: It would be helpful to say quickly in one sentence how the OSSE is set-up to make life "difficult" for the retrieval (includes plumes, realistic aerosols, etc) to make the results more meaningful.*
**Authors:** We added a sentence to the abstract that reads: "Since CO2M will not be launched until 2026, our study is based on simulated measurements over land surfaces from a comprehensive observing system simulation experiment (OSSE) that includes realistic meteorology, aerosols, surface BRDF (bidirectional reflectance distribution function), solar-induced chlorophyll fluorescence (SIF), and $CO_2$ and $CH_4$ concentrations."

***Reviewer 1:*** *Also, has any other retrieval method demonstrated they can meet the accuracy and precision requirements of CO2M, or is this the first? If it is the first, it's important to say so. Though it looks like RemoTAP also does, based on Lu et al 2022, is that also your read? If so then I guess say nothing...*
**Authors:** Lu et al. (2022) and Noël et al. (2024) both conclude that their retrieval meet the requirements when applied to their simulated CO2M data.

***Reviewer 1:*** *Abstract: I think it would be good to modify the abstract and conclusions to make it clear that you would have to re-train with real data once CO2M data are available, and that could change the storyline because of instrument artifacts, lack of sufficiently good training data (do you use TCCON, or a model, or...?). So while this is a solid proof-of-concept, we can only really believe the amazing results once you apply it to real data somehow.*
**Authors:** We added the following sentence to the end of the abstract and a similar sentence to the conclusions: "While the presented results are a solid proof of concept, the actual achievable quality can only be determined once NRG-CO2M is trained on real data, where it is confronted, e.g., with unknown instrument effects and systematic errors in the training truth."

***Reviewer 1:*** *Abstract: The sentence "We employ a hybrid learning*

*approach that combines advantages of simulation-based and measurement-based training data to ensure coverage of a wide range of XCO2 and XCH4 values making the training data also representative of future concentrations." Is important! But it downplays the excellent work you've done here. Even if your NN approach didn't work, this one thing is great and could be utilized by any researcher trying to do direct ML-retrievals of GHGs. Maybe change to "We created a novel hybrid learning approach...".*

**Authors:** We rephrased one sentence of the abstract, which now reads: "We created a novel hybrid learning approach that combines advantages of ...".

**Reviewer 1:** *You could also add a sentence like "This method could easily be applied by future researchers training MLbased GHG retrievals, to avoid this common problem." Or something to that effect. I think it's just important to highlight this contribution to the literature, in addition to your actual ML model.*

**Authors:** We added the following paragraph to the conclusions: "It should be noted that the method could be applied to other instruments and applications. In addition to generating representative training data, spectra could also be modified, e.g., to study the ability of a machine learning model to predict changes in its target variable.".

**Reviewer 1:** *Abstract: I think you should also add a sentence to the effect of "Our ML model also provides accurate estimates of both the noise-driven uncertainties and the averaging kernels of XCO2 and XCH4 for each sounding." This is an important aspect of your model; not all ML models do this.*

**Authors:** We added to the abstract: "In addition, NRG-CO2M also provides estimates of both the noise-driven uncertainties and the averaging kernels of XCO2 and XCH4 for each sounding."

**Reviewer 1:** *L43: BRDF -> surface BRDF*

**Authors:** Done.

**Reviewer 1:** *Fig1: For the love of god, please convince your CO2M colleagues to work in W m-2 µm-1 sr-1 units. We messed this up for OCO2/3. You can right this wrong.*

**Authors:** I'm afraid it's too late for that. In the paper we aimed at consistency with the MRD. Personally, I also like the SI units W m-2 µm-1 sr-1 more, but I think the instrument scientists are into photons per second, probably because this has to be multiplied with the quantum yield of the detectors to calculate the signal. It could have been worse: the number of photons could be given in Mol and I assume that the imperial measurement system could still provide some really nice area and length units :)

**Reviewer 1:** *Page6: How are clouds modeled in the radiative transfer? Do they come from CAMS? From where does the effective radius for water and ice come? Clouds were excluded in Noel et al (2024) for the FOCAL tests. It*

*seems like you are trying to include them here, so more details are welcome, since this is a specific difference to Noel et al.*

**Authors:** On page 6, we included: "For the SCIATRAN RT simulations, we used pressure, temperature, specific humidity, cloud ice content, cloud water content, and cloud fraction from the ECMWF ERA5 reanalysis (Hersbach et al., 2020). Since we focus mainly on cloud-free conditions, we used static cloud microphysical properties for convenience, representing spherical water droplets with a gamma particle size distribution with an effective radius of 12µm and fractal ice particles with an effective radius of 50µm (Fig. 3 of Reuter et al. (2010) shows the corresponding volume scattering functions)." As discussed in Sec. 2.5.2, we include some of the cloudy scenes in the training data, especially, those with little cloud optical depth. This is intended to make the prediction less sensitive to residual cloud contamination and mimics imperfect cloud clearing of the training data set. However, as mentioned in Sec. 3, all quality analyses are performed only for cloud-free scenes ("Since the CO2M mission requirements are defined for cloud-free conditions, we filtered the evaluation data accordingly.").

**Reviewer 1:** *Section 2.2. It's not clear how these uncertainties in dry-air column, temperature, CO2 profile etc are used. Are you saying that you stochastically apply these terms to the truth training data before you simulate the spectra? Or that you stochastically supply them as input to the NN predictions, so the NN doesn't have perfect knowledge of things like temperature profile, etc, when performing a retrieval on a given sounding? Please be clear. A flowchart might be helpful here. I think you ARE supplying these to the NN (you seem to say this in section 2.5) but please be explicit here. I think also saying WHY you need to supply this information is important.*

**Authors:** We added to Sect. 2.2: "'It should be noted that the input data for the RT simulations of the OSSE are free of noise. The main use of noise in our analyses is to generate realistically noisy training data. (Sect. 2.5).' Additionally, we added to Sect. 2.5 the explanation: "It is important that the training data set contains noise, as all input and target features will of course be subject to inherent uncertainties during later training with real CO2M data. In addition, the noise supports generalized learning and suppresses overfitting." Moreover, we added Fig. 1 of this document to the manuscript.

**Reviewer 1:** *Side note: I worry that you are telling your NN technique the answer by construction for each sounding, by supplying "truth data + gaussian noise" to it. It might be fine. But your "truth data + gaussian noise" for temperature, co2, surface pressure, etc, is not biased; there are no systematic errors. Instead, I would prefer that you had used a completely different model for your "prior information". For instance, CarbonTracker for CO2, MERRA-2 for Temperature, humidity, surface pressure, etc. Your hypothesis would be that it doesn't matter, but to me, that isn't clear.*

**Authors:** The purpose of the training data set is, of course, to teach the network the correct data and their relationships. Systematic errors bear the

[Figure]

Figure 1: *Baseline* ANN training setup on the example of XCO2, including the amount of noise added to the training features and to the target variable (Sect. 2.2) and the PCA components used (Sect. 2.4). When training with actual measured data in the future, the addition of noise will be omitted. Inst=Noise of instrument model; IL=input layer; HL=hidden layer; OL=output layer.

risk that incorrect relationships are learned, which leads to a degradation in the prediction quality. This risk is particularly present if biases in the target truth correlate with input features (e.g. systematically too high CO2 concentrations at high latitudes, or over bright surfaces). However, reliable information on such biases and their covariance statistics do not exist which is why we have not considered them and assumed Gaussian noise for convenience. At least our results become better comparable to those of Noël et al. (2024) who also used an unbiased a prior and an unbiased training truth for their machine learning based post processing bias correction. In order to make the reader aware of this point, we discuss in the introduction: "Obviously, such errors would have the potential to reduce the accuracy of the prediction, but a realistic estimate of the to be expected error patterns of the training truth is difficult and beyond the scope of this study."

**Reviewer 1:** *Near line 360. Feel free to add a contextual comment like: "For comparative purposes, the dry air column dependence for the operational OCO-2 XCO2 retrieval (v11.1) is roughly 85%, making it highly dependent on the accuracy of the prior meteorology, the prior surface elevation, and the instrument pointing (Jacobs et al., 2024, https://amt.copernicus.org/articles/17/1375/2024/)."*
**Authors:** Thanks, we added to section 3.3.1: "For comparison, the dry column dependence of the FOCAL CO2M XCO2 retrieval is 100% by design (Noël et al., 2024) and the dry column dependence of the operational OCO-2 XCO2 retrieval (v11.1) is approximately 85% (Jacobs et al., 2024)."

**Reviewer 1:** *Near line 420. I don't get why removing the NIR band doesn't increase the dependence on the dry air column to 100% ! Where is information on the dry column coming from? I guess from the fact that your prior co2 profiles are pretty good, so it can partially deduce the dry column from*

*the co2 bands alone?*
**Authors:** It cannot come from a too good a priori XCO2 because this would result in a larger dependency to the a priori. However, you probably meant the a priori profile shape. We agree that the CO2 profile shape has to be somewhat constraint in order to get dry column information from the CO2 bands. The a priori profile shape will contribute to this, but for the ANN, it would be sufficient that the CO2 profile shapes of the training data set vary not arbitrarily.

***Reviewer 1:*** *Also, regarding the increase in the dry column dependence when you remove MAP, from 6% to 16%. Typical surface pressure uncertainties are on order 1-2 hPa (or often even smaller). +- 2 hPa is 2/1000 roughly, and 10% of this is 2/10000. For a typical XCO2 of 400 ppm, this would induce an uncertainty of 0.08 ppm. This implies that removing MAP from CO2M which add an additional +- 0.08 ppm uncertainty to XCO2, due to errors in the prior surface pressure, relative to the with-MAP case. Which basically means that, according to your analysis, MAP really is not necessary. That's a pretty big conclusion that you are currently glossing over. Please address this directly in the manuscript. Presumably its due to some assumption you've made? FYI this also affects your interpretation in the conclusions (near 520), where you are implying that this is an important difference for the no-MAP case. It's really not, honestly. OCO-2/3 would kill to only have a 15% dependence on the dry air column, which leads to nearly negligible errors in the target gases.*
**Authors:** Within the conclusions, we modified the corresponding paragraph which now reads: "This had an apparently small effect on accuracy and precision, which is not consistent with the results of Lu et al. (2022), whose retrieval method became significantly less accurate under these conditions. We can only speculate about possible reasons for this. i) We use a different aerosol microphysical model, which is consistent with the MACC aerosol model, but is less complex than the one used by Lu et al. (2022). ii) Their CO2I-only retrieval method is fundamentally different from ours and also from FOCAL, which may result in different sensitivities to aerosol-induced biases. In this context, it should be noted that our CO2I-only results are in good agreement with those of Noël et al. (2024), suggesting that it may be possible to meet the CO2M mission requirements without using MAP. iii) The statistics computed by Lu et al. (2022) to quantify the systematic and stochastic errors differ from those computed by us. However, we observe that the dependence of the XCO2 prediction on the dry column increases when MAP is not used, which may introduce systematic errors of the order of 0.1 ppm in reality when perfect knowledge of the dry column cannot be expected."

***Reviewer 1:*** *Near Line 470, and Figures 10+11. Can't you plot the AK-corrected Truth minus Prediction, instead of straight truth − prediction? You should! I \*always\* do this in my OSSE experiments, it is important. It would also show if your hypothesis is correct on the source of this hotspot in the difference plot of figure 11. In fact a comparison of these two plots (with and without AK-correction) would be very illuminating. Your statement on using*

*the true profiles as prior comes close to accomplishing this, but is not nearly as powerful. Plus, you are expecting modelers to make the AK correction; therefore I think It's important to set a good example and do the same, and show the effect when you don't.*

**Authors:** When AKs are taken into account, the difference between modeled and true XCO2 is

$$\Delta X = \hat{X} - \sum w_i [c_i^{apr} + A_i(c_i^{mod} - c_i^{apr})] \tag{1}$$

where $\hat{X}$ is the retrieved XCO2, $w$ is the weighting of layer $i$, $c^{apr}$ is the a priori profile, and $c^{mod}$ is the model profile. Most of our analyses have been done with an a priori equal to the truth, i.e. MACC. In this case, $c^{apr}$ becomes $c^{mod}$, so that the difference between retrieved and true XCO2 becomes

$$\Delta X = \hat{X} - X^{mod}. \tag{2}$$

This is the quantity we analyze to assess the systematic errors, as shown in Fig. 6 and 7. This means using the truth as a prior has the advantage that all deviations from the truth can be directly attributed to retrieval deficiencies without explicitly accounting for the AKs. However, it has the disadvantage that it rewards retrievals that put little weight on the measurement and much weight on the a priori. In order to demonstrate, that this is not the case here, we performed the anaylses of the Berlin scene on purpose with a constant a priori so that it is clear that the retrieved XCO2 variability only comes from the measurement but not the a priori.

If we understand the comment correctly, you are suggesting to also show results for the Berlin scene with AKs applied. In this case, $\Delta$XCO2 would become

$$\Delta X = X^{con} - \sum w_i [c_i^{con} + A_i(c_i^{mod} - c_i^{con})] \tag{3}$$

where $X^{con}$ is the retrieved XCO2 using the constant a priori and $c^{con}$ is the constant a priori profile. Using the AKs, we can compute $X^{con}$ from the retrieval result $\hat{X}$ obtained using the model as a priori:

$$X^{con} = \hat{X} - \sum w_i (1 - A_i)(c_i^{con} - c_i^{mod}) \tag{4}$$

so that

$$\Delta X = \hat{X} - X^{mod}. \tag{5}$$

This equation is the same as Eq. 1 which means, that the difference between the prediction using the truth as a priori and the model equals the difference between the prediction using the constant a priori and the model with AKs applied. In other words, instead of applying the AKs to the model, we can also use the truth as a priori (as, e.g., in Fig. 6 and 7). We added the corresponding figures to the appendix of the manuscript (see Fig. 2 and 3 of this document).

[Figure]

Figure 2: As Fig. 11, but using the true $CO_2$ concentration profiles as a prior instead of their scene-wide average.

[Figure]

Figure 3: As Fig. 12, but using the true $CH_4$ concentration profiles as a prior instead of their scene-wide average.

**Reviewer 1:** *L502: short correlation length parts -> or short correlation length parts*

**Authors:** Done.

**Reviewer 1:** *I think the conclusions section really needs a paragraph on what it would take to "operationalize" this algorithm for real satellite data. Presumably you would train it on observed spectra, along with your method to extend it to larger truth values of XCH4 and XCO2? What would you use for the training truth: TCCON, Models, something else? Would your methods to get at the AK and posterior Xgas uncertainties still work? Would you have any reason to expect worse performance?*

**Authors:** We replaced the last paragraphs of the conclusions, which now reads:

[revised manuscript text omitted]

---

## Author Comment (AC2)

First of all, we thank reviewer 2 for his/her effort in carefully reviewing our manuscript and his/her constructive comments.

**Point-by-point answers to the comments of reviewer 2**

**General comments**

**Reviewer 2:** *For example, to retrieve the XCO2 and XCH4, the instrumental model is very important. In this manuscript, only the random noise is assessed. The authors should concern the other parameters at least the uncertainly of instrumental line shape function and its wavelength depended response. In addition, the authors were used the actual space-based observation data such as OCO-2 during the FOCAL development. To evaluate the new NRG-CO2M algorithm with actual space-based observation data with realistic uncertainty is also important and informative. However, the authors are only focused the simulation-based dataset. I understand the CO2M will not be launched until 2026. The authors should be considered the evaluation plan with the updated instrumental model data and the realistic characterization error, and these impact on the NRG-CO2M processing. Furthermore, the application for the actual space-based observation dataset, currently available dataset, is also informative and productive for the evaluation purpose. The authors should be considered the evaluation plan for the NRG-CO2M with currently available observation dataset. I recommend the authors will add the sentences and clarify for some of unclear sentences. For these reasons, I recommend this paper for publication with minor changes to the technical content.*
**Authors:** The reviewer raises two general points related to the instrumental model used and the fact that our study is based only on simulations and not on actual measurements from existing satellite instruments. Since both points are also raised in the section "specific comments", we'll address them in that section.

**Specific comments**

**Reviewer 2:** *Page 1, line 13: Spell out first for "NRG-CO2M". -> Neural networks for Remote sensing of Greenhouse gases from CO2M (NRG-CO2M)*
**Authors:** Done.

**Reviewer 2:** *Page 1, line 19: The definition of "spatio-temporal systematic errors" is unclear. The authors should add the definition or more clear explanation for the condition.*
**Authors:** We now define the term "spatio-temporal systematic errors" earlier in the abstract: "According to the CO2M mission requirements, the spatial and temporal variability of the systematic errors (or spatio-temporal systematic

errors) of XCO2 and XCH4 ...".

**Reviewer 2:** *Page 2, line 39: add the ",", between "5ppb" and "respectively".*
**Authors:** Done.

**Reviewer 2:** *Page 2, line 41: Spell out first for "CO2I".*
**Authors:** Done.

**Reviewer 2:** *Page 2, line 42: Spell out first for "MAP".*
**Authors:** Done.

**Reviewer 2:** *Page 2, line 42: Spell out first for "BRDF".*
**Authors:** Done.

**Reviewer 2:** *Page 2, line 43: Spell out first for "CLIM".*
**Authors:** Done.

**Reviewer 2:** *Page 2, line 44: XCO2 or XCH4 -> XCO2 and/or XCH4*
**Authors:** Done.

**Reviewer 2:** *Page 2, line 47: 2017b,a -> 2017 a, b*
**Authors:** Done.

**Reviewer 2:** *Page 2, line 49: Spell out "EUMETSAT".*
**Authors:** According to the AMT author guidelines, abbreviations that are better known than their full form need not be defined, which is the case here.

**Reviewer 2:** *Page 2, line 57: The meaning of "3D effects" is unclear. The authors should add the explanation.*
**Authors:** The radiative transfer (RT) models used in atmospheric greenhouse gas retrievals are so-called 1D RT models, because they consider changes of the atmospheric properties only in one dimension. I.e., all properties change only with height. As a consequence, photon transport between neighboring columns with different properties is not possible so that atmospheric columns can be considered independent. However, in reality such photon transport happens which results in inaccuracies of 1D RT models. Especially near cloud edges these inaccuracies can become important. RT-models that are able to account for atmospheres with varying properties in three dimensions are called 3D RT models. As they are usually computationally more expensive and as 3D properties of the atmosphere are often not known, they are not used in operational satellite greenhouse gas retrievals. Whenever limiting changes in the atmospheric properties to 1D results in inaccuracies, we speak of 3D effects.
Since we use the term "3D effect" only as a keyword for further reading and as one of several examples, and since the term is common in the context of RT modeling, we would like to avoid a more detailed description in the paper.

Otherwise, we could also remove the term.

**Reviewer 2:** *Page 3, line 61: Spell out first for "OCO-2".*
**Authors:** Done.

**Reviewer 2:** *Page 3, line 62: Spell out first for "GOSAT".*
**Authors:** Done.

**Reviewer 2:** *Page 3, line 77: the meaning of "meteorology and angles" are unclear. The authors should add the explanation.*
**Authors:** We rephrased to " meteorological profiles, observation angles". In the context of page 3, line 77, these are only unspecific examples. A detailed description of the input features to the MLPs trained by us is given later in the paper.

**Reviewer 2:** *Page 3, line 83: Krasnopolsky and Schiller (2003). -> (Krasnopolsky and Schiller, 2003).*
**Authors:** Done.

**Reviewer 2:** *Page 4, line 116: Spell out first for "OSSE".*
**Authors:** Starting on P4 L115, the manuscript reads: "... is based on simulated measurements from an extensive observing system simulation experiment (OSSE), which is a refinement of ..."

**Reviewer 2:** *Page 4, line 116: In the previous works, the authors were developed FOCAL full physics algorithm. During the development phase of FOCAL, the authors are actually used the space-based observation data such as OCO-2 and GOSAT. To evaluate the new NRG-CO2M algorithm with actual space-based observation data is quite realistic and import. However, the authors are only focused the simulation-based dataset. So, the authors should be considered the evaluation plan with actual space-based observation dataset or current limitations.*
**Authors:** We agree with the reviewer that the application to real data from OCO-2, GOSAT, or GOSAT-2 could provide interesting additional results. However, due to the complexity of the work required, this would change the FOCUS of the paper significantly which is to use simulations to develop a technique to modify measured spectra in such a way that they can be used as representative training data for hitherto unprecedented atmospheric conditions and that MLP-based methods are able to fulfill the CO2M mission requirements under suitable conditions. In this respect, the results shown should rather be interpreted as a proof-of-concept (see also reviewer 1). Accordingly, we discuss in the revised manuscript: "In the analysis of real data, several effects, the detailed investigation of which is beyond the scope of this paper, may lead to somewhat degraded retrieval quality. These include unknown systematic errors in the training truth, a priori, and met profiles, non-ideal sampling of the training data set, and potential instrument or RT features that are not well

approximated by our spectrum modification method. Therefore, the actual retrieval quality achievable can only be determined after NRG-CO2M has been trained on and applied to real data." We would also like to point out that due to the differences between the OCO-2 and CO2M instruments, perfect transferability of the results would not be guaranteed.

**Reviewer 2:** *Page 5, line 154: What is the instrumental line shape model? It also has several uncertainties. It is not clear how to take account spectrally depended uncertainties. The authors should add the explanation.*
**Authors:** As our OSSE setup including the instrument model is widely adapted from Noël et al. (2024), we only briefly describe the setup. However, we now added to Sect. 2.1: "The simulated main instrument CO2I consists of four imaging spectrometers for the wavelength ranges 405 nm–490 nm (VIS, $NO_2$), 747 nm–773 nm (NIR, $O_2$), 1590 nm–1675 nm (SWIR-1, $CO_2$ and $CH_4$) and 1990 nm–2095 nm (SWIR-2, $CO_2$) having spectral resolutions of 0.6 nm, 0.12 nm, 0.3 nm and 0.35 nm, respectively. In line with currently available information about CO2I, the instrument line shape functions are assumed to be Gaussian with full width at half maximum, corresponding to the respective spectral resolution."

**Reviewer 2:** *Page 7, line 193: How to consider the bias in a priori? Especially in the future prediction, not only a standard deviation but also the bias has to be considered. The authors should add the explanation.*
**Authors:** Biases in the a priori or the training truth have the potential to introduce biases in the prediction. In case of the a priori this is usually less problematic because the influence of the a priori is reduced when applying the averaging kernels when using the prediction for emission estimation. However, systematic errors in the training truth bear the risk that incorrect relationships are learned, which is particularly possible if biases in the training truth correlate with input features. (e.g. systematically too high CO2 concentrations at high latitudes, or over bright surfaces). Unfortunately, reliable information on such biases and their covariance statistics do not exist which is why we have not considered them and assumed Gaussian noise for convenience. At least our results become better comparable to those of Noël et al. (2024) who also used an unbiased a prior and an unbiased training truth for their machine learning based post processing bias correction. In order to make the reader aware of this point, we discuss in the introduction: "Obviously, such errors would have the potential to reduce the accuracy of the prediction, but a realistic estimate of the to be expected error patterns of the training truth is difficult and beyond the scope of this study."

**Reviewer 2:** *Page 18, Figure 5: How is the slope? It seems that the linearity can be directly estimated from this analysis. However, it is not mentioned in the text.*
**Authors:** We updated Fig. 5 of the manuscript (Fig. 1 in this document) which now also includes the results of a linear regression.

[Figure]

Figure 1: Comparison of postprocessed predicted XCO2 (a) and XCH4 (b) with corresponding true values for noise-free 2020 subset input data. $\Delta$ represents the average prediction error (prediction minus true), $\sigma$ the standard deviation of the prediction error, and $\Sigma$ the total number of soundings. The figure also contains the results of a linear regression.

**References**

Noël, S., Buchwitz, M., Hilker, M., Reuter, M., Weimer, M., Bovensmann, H., Burrows, J. P., Bösch, H., and Lang, R.: Greenhouse gas retrievals for the CO2M mission using the FOCAL method: first performance estimates, Atmospheric Measurement Techniques, 17, 2317–2334, https://doi.org/10.5194/amt-17-2317-2024, 2024.